# Phase-Inserted Fiber Gratings and Their Applications to Optical Filtering, Optical Signal Processing, and Optical Sensing: Review

**Chengliang Zhu** [1,2,3], **Lei Wang** [1] **and Hongpu Li** [4,*]

1 College of Information Science and Engineering, Northeastern University, Shenyang 110004, China; zhuchengliang@neuq.edu.cn (C.Z.); 2071965@stu.neu.edu.cn (L.W.)
2 Hebei Key Laboratory of Micro-Nano Precision Optical Sensing and Measurement Technology, Qinhuangdao 066004, China
3 State Key Laboratory for Integrated Automation of Process Industry, Shenyang 110004, China
4 Graduate School of the Engineering, Shizuoka University, 3-5-1 Johoku, Naka-Ku, Hamamatsu 432-8561, Japan
* Correspondence: ri.kofu@shizuoka.ac.jp

**Abstract:** Phase-inserted fiber gratings (PI-FGs) refer to those gratings where there exist a number of the phase-shifts (spatial spacing) among different sections (or local periods) of the gratings themselves. All the PI-FGs developed to date can mainly be divided into three categories: phase-shifted gratings, phase-only sampled gratings, and phase-modulated gratings, of which the utilized gratings could be either the Bragg ones (FBGs) or the long-period ones (LPGs). As results of the proposed the PI-FGs where the numbers, quantities, and positions of the inserted phases along the fiber direction are optimally selected, PI-FGs have already been designed and used as various complex filters such as the ultra-narrow filters, the triangular (edge) filters, the high channel-count filters, and the flat-top band-pass/band-stop filters, which, however, are extremely difficult or even impossible to be realized by using the ordinary fiber gratings. In this paper, we have briefly but fully reviewed the past and recent advances on PI-FGs, in which the principles and design methods, the corresponding fabrication techniques, and applications of the different PI-FGs to the fields of optical filtering, optical signal processing, and optical sensing, etc., have been highlighted.

**Keywords:** phase-shifted gratings; phase-only sampled gratings; phase-modulated gratings; fiber Bragg gratings; long-period fiber gratings; helical long-period fiber gratings; all-optical signal processing devices; optical fiber sensors

## 1. Introduction

Fiber grating refers to a piece of fiber where there exists a periodic index modulation in either the core or the cladding region along the fiber axis direction. Since 1978, when Hill et al., first invented the fiber grating [1], countless kinds of fiber gratings have been proposed and demonstrated to date, which, however, can mainly be divided into two categories: fiber Bragg gratings (FBGs) and long-period fiber gratings (LPGs) [1–184], of which the phase-inserted fiber gratings (PI-FGs), including the phase-inserted FBGs and the phase-inserted LPGs, refer to the gratings where there exist a single or a large number of the phase-shifts among the local sections (or periods) of the gratings. All the PI-FGs proposed and demonstrated to date can mainly be divided into three categories, i.e., the phase-shifted fiber gratings [6–102,137–174], the phase-only sampled fiber gratings [104–126,175–178], and the phase-modulated fiber gratings [127–135,180–184]. For an intuitive view, the structure diagrams, phase distributions, and index modulations of both the uniform grating and the PI-FGs are schematically depicted in Figure 1, respectively, where Figure 1a corresponds to case of a uniform fiber grating. As is shown, in this case the index modulation distribution

is of a cosine-like one with a period of $\Lambda_0$, while the local phases remain a constant through whole length of the grating. Figure 1b corresponds to the case of a phase-shifted fiber grating, where there exist one or several phase jumps in the phase distribution of the grating itself. Such a phase jump is also called the phase-shift, which in general can be equivalently realized by inserting an additional blank region $D_\varphi$ in the distribution of index modulation, as shown in Figure 1b. Figure 1c shows the case of a phase-only sampled fiber grating, where the local phase of the grating is modulated by a periodic function, generally called the phase-only sampling function, and the period of such a phase function is much larger than that of grating itself. The phases of the sampling function could be discrete or continuous ones, and the continuous one can be discretely divided and coded into each local period of the seed grating. The phase-only sampled fiber grating is commonly used to generate multiple channels in the spectrum. Figure 1d corresponds to the case of a phase-modulated fiber grating, where the local phase of the grating is not a periodic one but an either linear or nonlinear function of the grating's position $z$. The most typical example of such kinds of gratings could be the linearly chirped FBG, where linear change of the period in terms of the position in fact can be equivalently expressed as the FBG's local phase but with a quadratic relation with the position $z$ [5]. Similarly, phase distributions of such gratings could be either discrete or continuous ones, and in fabrication, the continuous phase can be discretely divided and encoded into each local period of grating itself.

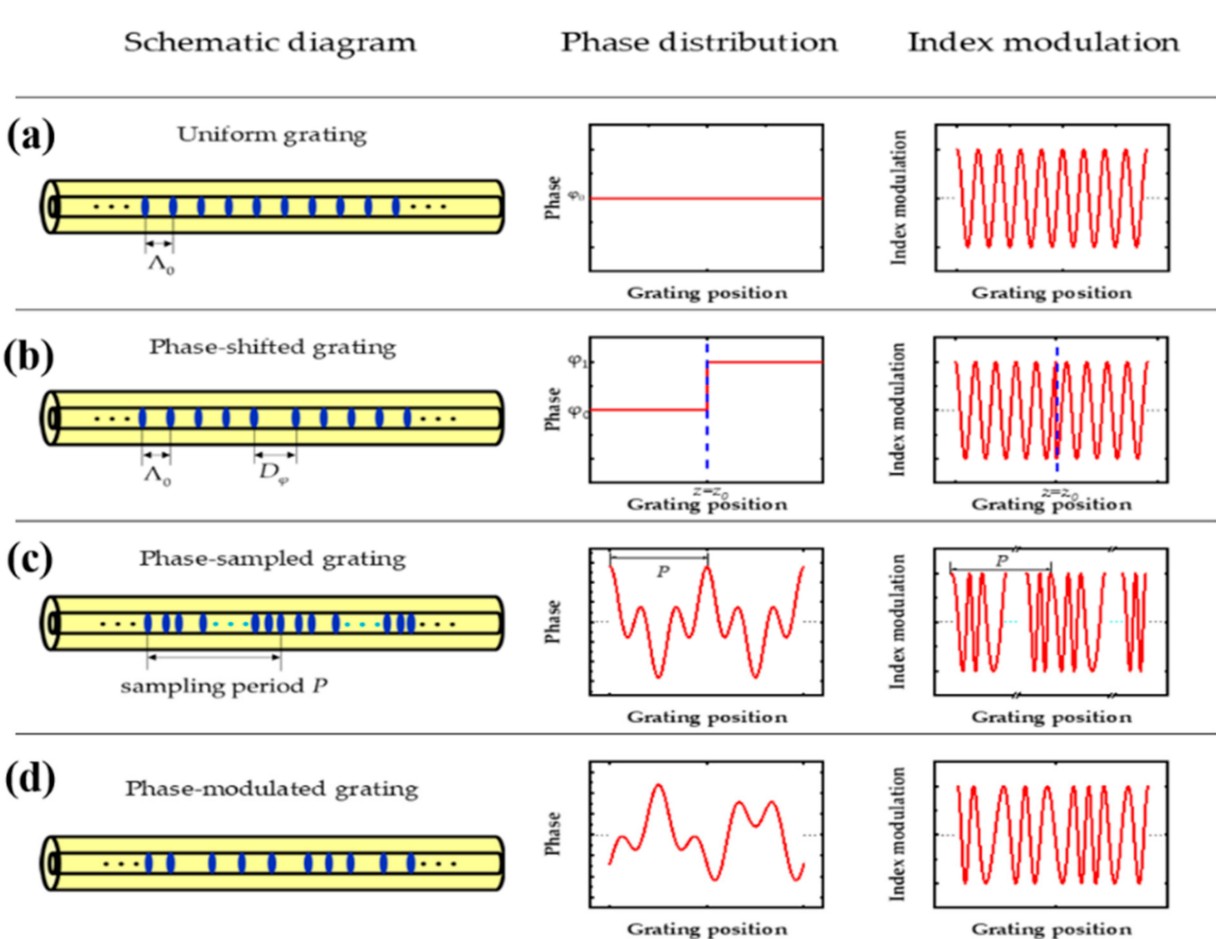

**Figure 1.** Schematic diagrams, phase distributions, and index-modulations of (**a**) the conventional, (**b**) the phase-shifted, (**c**) the phase-only sampled, and (**d**) the phase-modulated fiber gratings.

To date, as results of the proposed PI-FGs where the numbers, quantities, and positions of the inserted phase-shifts along the fiber direction are suitably selected with a high-degree

of freedom, PI-FGs have already been designed and used as various complex filters such as ultra-narrow filters, triangular (edge) filters, high channel-count filters, and flat-top band-pass/band-stop filters, which, however, are extremely difficult or even impossible to be realized by using the conventional fiber gratings. Moreover, it must be pointed out that the strength, i.e., the maximum index-modulation of PI-FGs, remains a constant within whole length of the grating, which could considerably facilitate the grating fabrication itself owing to the elimination of the complex apodization in grating's amplitude.

In this paper, we briefly but fully review the past and recent advances on PI-FGs, in which the principles, the design methods, the fabrication techniques, and versatile applications of the PI-FGs especially in the fields of optical communications, optical signal processing, and optical sensors are highlighted. In Section 2, advances on FBG-based PI-FGs and their applications are introduced and discussed, whereas in Section 3, advances on the LPG-based PI-FGs and their applications are introduced and discussed. In Section 4, the conclusions and the prospects for the works on PI-FGs-based devices are introduced and discussed.

## 2. Phase-Inserted Fiber Bragg Grating

FBG refers to a fiber grating where the period of the index modulation is nearly the same order as the light wavelength. As a result, the mode-coupling resonantly occurs only between the forward-propagating core mode and the backward-propagating core mode, behaving exactly like a bulk Bragg grating, which makes the FBG a good candidate as a narrow band-pass filter (with a bandwidth ~0.1 nm) but operating in the reflection [3]. Nowadays, FBGs have been widely used as fiber sensors, wavelength division multiplexer (WDM) demultiplexers, add/drop filters, and dispersion compensators, etc. [2–6]. On the other hand, it is rather difficult or even impossible to flexibly and precisely control the bandwidth and the spectral profile of the ordinary FBG, especially for the ones with either an ultra-narrow bandwidth or a triangular envelop, or a rectangular-type envelop, or a high channel count in spectrum, etc., which, however, is essential to the FBG-based devices whenever they are practically used in the fields of fiber communications, optical information processing, and fiber sensing, etc. Phase-inserted FBGs are excellent candidates that enable overcoming all the issues mentioned above.

### 2.1. Phase-Shifted FBGs (PS-FBGs) and Their Applications

2.1.1. Principle and Fabrication Methods of the PS-FBG

In general, the refractive index-modulation of FBG can be expressed as

$$\Delta n(z) = \text{Re}\left\{\frac{\Delta n_1(z)}{2} \exp\left(i\frac{2\pi}{\Lambda_0}z + i\phi_g(z)\right)\right\} \tag{1}$$

where Re represents the real part of a complex number. $z$ represents the position in the grating, $\Delta n_1(z)$ represents the maximum index modulation of the grating, and $\Lambda_0$ and $\phi_g(z)$ represent the central period and local phase of the grating, respectively. Note that for convenience, the DC part of the index-modulation has been ignored in Equation (1).

PS-FBG is generally realized by inserting a single or several abrupt phases (called the phase-shifts) into specific positions of the FBG. When the phase-shift $\theta_j (j = 1, 2, \ldots, N)$ is inserted at position $z_j$, the index modulation of such FBG then can be mathematically expressed as

$$\Delta n_p(z) = \text{Re}\left\{\frac{\Delta n_1(z)}{2} \exp\left(i\frac{2\pi}{\Lambda_0}z + i\phi_g(z) + i\sum_{j}^{N}\theta_j \cdot q(z - z_j)\right)\right\} \tag{2}$$

where $\theta_j \cdot q(z - z_j)$ and $N$ represent the magnitude and number of inserted phases in the grating, respectively. $q(z - z_j)$ represents a step function, i.e., $q(z - z_j) = 0$ when $z < z_j$ and $q(z - z_j) = 1$ when $z \geq z_j$. Once if the phase-shift $\theta_j \cdot q(z - z_j)$ and other parameters

including the maximum index modulation, the grating period, local phases, and total length of the FBG are determined, the reflection/transmission spectrum of the PS-FBG can be calculated by using the transfer matrix method [5]. Figure 2 shows the transmission spectrum of one typical PS-FBG, where a $\pi$ phase-shift is inserted at the middle of a uniform FBG [7]. From this figure, it can be seen that due to the insertion of the $\pi$ phase-shift, the loss band of the original FBG (i.e., the FBG without phase-shift) is split into two, and accordingly in the transmission, an ultra-narrow band-pass spectrum can be obtained at the central wavelength.

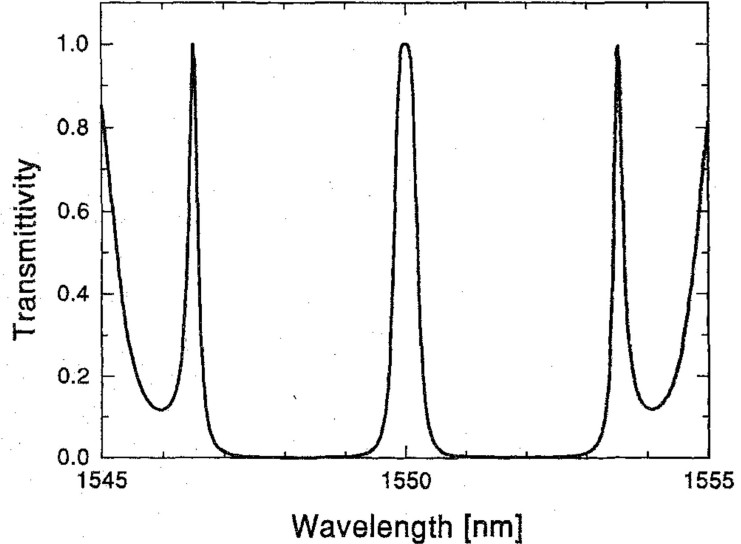

**Figure 2.** Transmission spectrum of a PS-FBG where a phase-shift of $\pi$ is inserted at middle of the FBG. Adapted with permission from ref. [7]. © [2022] IEEE.

In order to practically fabricate the PS-FBG, various methods have been proposed and demonstrated so far. All of the methods can be divided into two types: the permanent and the temporary phase insertion methods. For the former one, the required phase-shift is permanently originated and inserted at a definite position of the FBG, whereas for the latter one, the phase-shift is temporarily originated and inserted into FBG, i.e., the magnitude as well as the position of the inserted phase-shift can be changed as one wills.

Figure 3 shows several fabrication methods particularly developed for permanent insertion of the phase-shift into FBG, which include the phase-shift phase mask method [9], the post-processing methods [10–15], and the femtosecond laser-based direct writing techniques [16,17]. Among these methods, the phase-shift phase mask technique was firstly proposed and demonstrated by Kashyap et al., as shown in Figure 3a, where the desired phase-shift was pre-inscribed in a phase mask and, as a result, the phase-shift as well as the FBG itself are produced at the same time. The phase-shift phase mask method provides us a robust means that enables the fabrication of the PS-FBG with a higher quality and repeatability than those of the other methods proposed so far. However, such a technique undergoes two inevitable shortcomings, i.e., the high cost for a phase mask and less of the flexibility in fabrication than any other kinds PS-FBGs once the phase mask is fabricated.

The post-processing technique is another effective way widely used to insert a phase-shift permanently in FBG, which also include the ones based on the ultraviolet (UV) irradiation [10], the $CO_2$ laser irradiation [11], the femtosecond laser-irradiation [12], the arc-discharge [13], and the chemical etching [14,15], respectively. Among these, Canning et al., firstly proposed and demonstrated the post-UV-irradiation method as shown in Figure 3b, where the inserted phase-shift was accumulated and equivalently obtained by changing the effective index of the FBG within a central small region, which was realized by letting such a small part of the FBG exposure directly by the UV laser but without the

phase mask. As a result, a phase-shift FBG was successfully obtained and inserted at the central part of the FBG [10].

Instead of the UV laser, the $CO_2$ laser, the femtosecond laser, and the arc-discharge have also been used to form and insert a post-processing phase-shift into a fabricated FBG; typical examples of them are shown in Figure 3c–e, respectively, where the principles to produce the phase-shift are almost the same as those of the post-UV irradiation, i.e., the phase-shifts are produced due to a change in the effective index induced in local region of the FBG [11–13]. The magnitude of the inserted phase-shift can be changed by adjusting the width of the irradiation region and the irradiation flux as well.

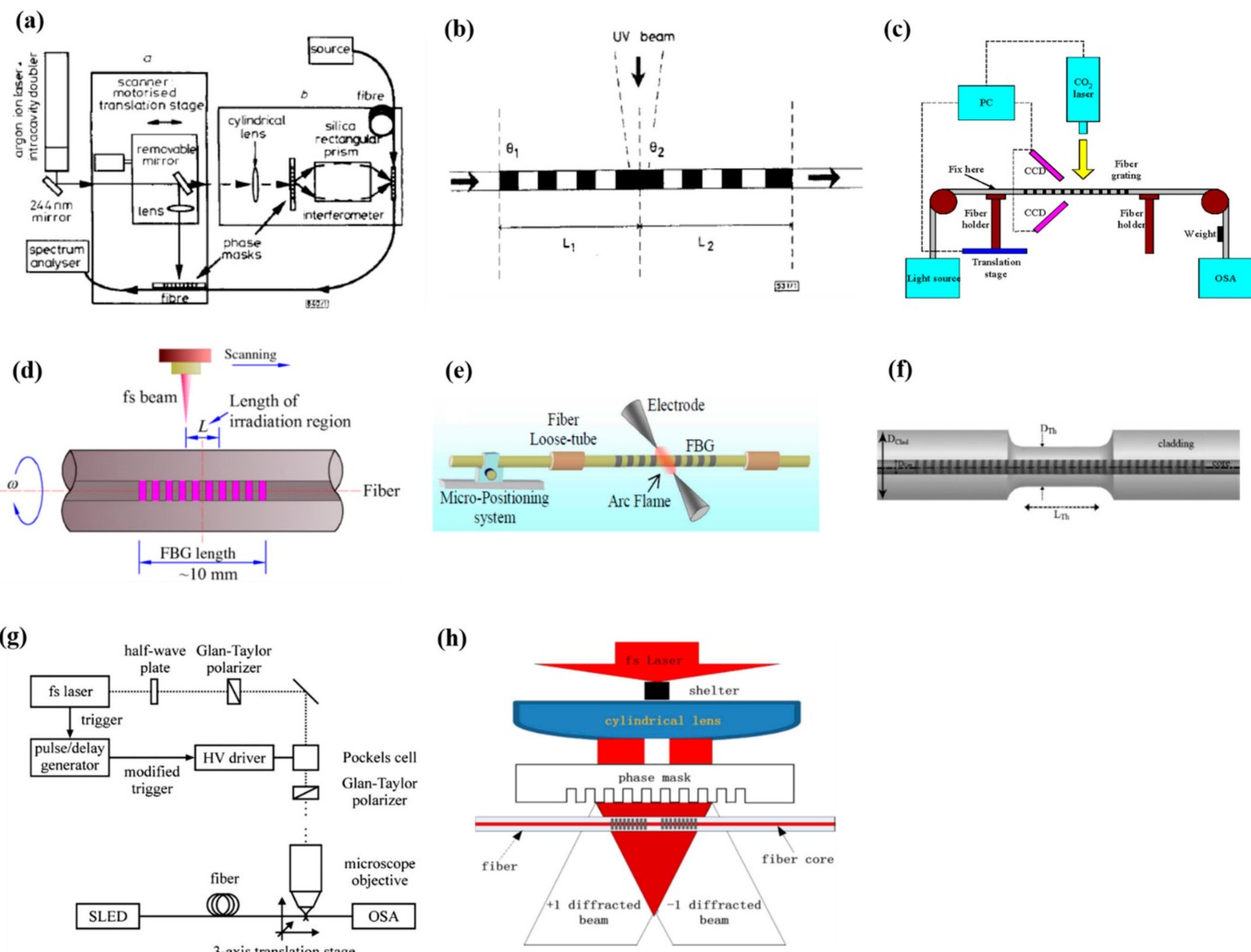

**Figure 3.** Fabrication methods for insertion of the permanent phase-shift into FBG. (**a**) Phase-shift phase mask method [9]. (**b**) The approach to erase the local part of the grating by using the UV light [10]. Post-processing techniques based on (**c**) the $CO_2$ laser irradiation [11], (**d**) femtosecond laser irradiation [12], and (**e**) electric arc discharge [13]. (**f**) Post-processing technique based on the chemical etching method. Adapted with permission from ref. [15]. (**g**) Point-by-point femtosecond laser writing technique [16]. (**h**) Shielded phase mask-based femtosecond laser writing technique. Adapted with permission from ref. [19].

Besides those post-irradiation methods mentioned above, Cusano et al. [15] proposed and demonstrated another post-processing method, namely the post-etching method as shown in Figure 3f, where the accumulated phase-shift introduced in the central region of the pre-fabricated FBG was obtained by partially and locally etching the cladding layer along the grating direction. The principle of this method relies on the fact that the effective

index of the core is dependent on the effective indices of the cladding and the surrounding material once if the cladding layer is thin enough [14]. Therefore, the magnitude of the phase-shift produced by using this method is mainly dependent on three parameters, i.e., the length and depth of the etched region and the surrounding refractive index.

Unlike all the post-processing methods mentioned above, the infrared femtosecond (fs) laser writing methods have recently been proposed and demonstrated to fabricate PS-FBGs [16–21]. Of these, Burgmeier et al., had firstly reported and demonstrated a PS-FBG based on the point-by-point (or line-by-line) technique as shown in Figure 3g [16], where the FBG was point-by-point inscribed in a fiber by using a femtosecond laser while the phase-shift inserted into the FBG was realized just by setting an appropriate line gap at the required local position of the grating, i.e., the FBG's fabrication and the phase-shift insertion are realized simultaneously without any post-processing operations. However, an extremely high precision for the fabrication setups and high quality for the used laser beam are strictly required, which would inevitably restrain this technique from the industry application for mass production. As an alternative, Du et al., have proposed and demonstrated a simple PS-FBG writing technique [19], shown in Figure 3h, where a uniform phase mask and an additional beam shelter (blocker) are specially utilized. During the fabrication, a central part of the fs laser beam is blocked by the shelter. As a result, there exists a blank (un-exposure) region in the central region of the FBG, which is equivalent to a phase-shift inserted in between two identical FBGs located on both sides of the blank region. However, such a newly produced phase-shift strictly depends on the length of the blank region formed in FBG; in order to precisely obtain the phase-shift, much more effort in the calibrations of the practically obtained phase-shift and optimal adjustments of the shelter, in both the length and position with a resolution of sub-micrometers, then becomes necessity.

In addition to the cases that the phase-shifts are permanently inserted into the FBG, the phase-shifts can also be temporally formed and inserted in an ordinary FBG. Figure 4 shows six typical examples for temporal generation a phase-shift in an FBG, which include thermal method [22–27], mechanical tensile method [28–34], magnetostriction method [35,36], and micro-channel methods [37,38], etc. In most of the above methods, the phase-shift is obtained by producing a local defect on a fabricated FBG, which could be either the thermally induced, the mechanically induced, the piezo-induced, the electric arc discharge-induced, or the magnetostriction-induced ones. Of these, Janos et al., first proposed and demonstrated a thermally induced a tunable phase-shift in uniform FBG as shown in Figure 4a, where the insertion of the phase-shift was realized by using a NiCr wire heater [23]. The mechanism to produce the phase-shift is based on the thermal expansion and thermal-optical effect of the fiber, i.e., the length and the refractive index of the heated fiber will change a little, and as a result, a phase-shift could be introduced within the thermal region. The magnitude of the inserted phase-shift can easily be changed by adjusting the voltage applied on both ends of the wire heater. The thermal method with either single or multiple NiCr wires have also been used to produce the wide-band tunable filter and the comb filter, etc. [24–27]. The proposed method has the advantages of a simple structure and a consecutive tunability in the magnitude of the phase-shift. However, due to the inherent thermal diffusion effect, it is rather difficult to obtain the phase-shift precisely at a definite local point. In addition to the wire heater methods, Ahuja et al., have proposed and demonstrated another thermal method to produce a phase-shift in FBG as shown in Figure 4b [27], where instead of the NiCr wire heater, a thin metal film deposited on the surface of the FBG was used as a local heater. The proposed thin film method has the advantages of more compactness and less power consumptions than the bulk wire heater method demonstrated in [23].

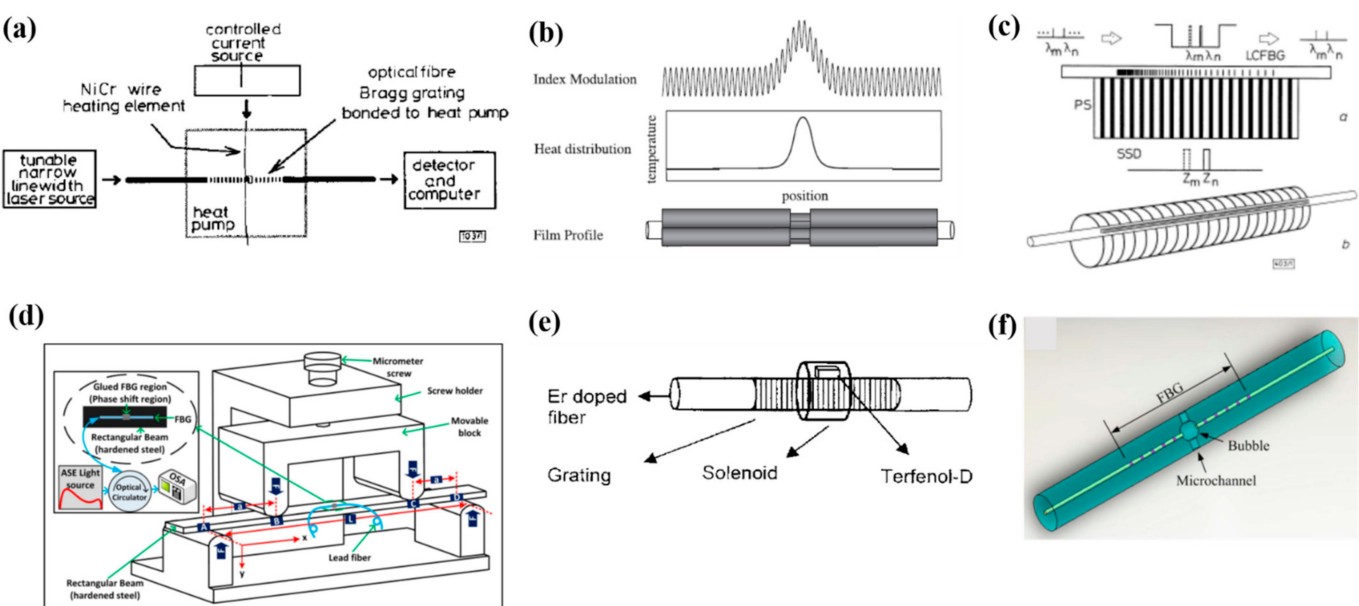

**Figure 4.** Fabrication methods for insertion of a temporal phase-shift in FBG. (**a**) NiCr wire heating method [23]. (**b**) Thin metal film heating method. Adapted with permission from ref. [27]. (**c**) Piezoelectric transducer method [28]. (**d**) Mechanical pressure method. Adapted with permission from ref. [34]. (**e**) Magnetostrictive transducer technique [35]. (**f**) Femtosecond laser-based microchannel method [37].

Besides those thermal methods described above, the methods based on the mechanical stretching and mechanical bending have also been exploited to insert phase-shifts in FBGs [28–34]. Xu et al., firstly demonstrated a linearly chirped FBG but with tunable multiple-phase-shifts, which was realized by using the piezoelectric transducer (PZT) technique [28]. Figure 4c shows a schematic diagram of such a method, where a PZT stack is intensively glued to the surface of an FBG. When an external voltage is applied to the PZT, whose length will be changed a little. Within the same region as the PZT, the glued FBG will undergo the same change in its length, which in return means that an equivalent phase-shift will be inserted into the FBG. Such method enables to insert a temporary phase with a high precision. Moreover, in addition to the single phase-shift, multiple phase-shifts can easily be inserted in an FBG as long as the multiple PZTs are utilized and operated simultaneously [31]. As an alternative to the PZT ones, Falah et al., have proposed and demonstrated another mechanical method, shown in Figure 4d, where the phase-shift insertion and the magnitude tuning was realized by bending the local part of the FBG through the mechanical parts. The proposed method enables to provide a simple and precise control of the phase-shift with either positive or negative magnitude (with a range of $-\pi$ to $\pi$).

Besides the above ones, the methods based on the magnetostrictive transducer have also been proposed and demonstrated [35,36]. In 2005, Perez-Millan et al., reported and demonstrated an active Q-switched distributed feedback, erbium-doped fiber laser, in which a phase-shifted FBG was particularly used as a Q-switch (cavity-loss control) component, as shown in Figure 4e, where the dynamic phase-shift was induced by a small magnetostrictive actuator that could be electronically controlled by a small solenoid. The proposed method has the advantages of the fast response time and low power consumption as well. In 2013, Liao et al., proposed another permanent phase-shift insertion method, shown in Figure 4f [37], where an inner bubble structure (microchannel) was post-created in an FBG by using a femtosecond laser. With this method, a phase-shift can be equivalent created at a position of bubble and magnitude of such a phase-shift can be adjusted by injecting the liquids with different refractive indices. More details about the technique and fabrication procedures can be found in [37].

Last but not the least, besides the methods shown in Figures 3 and 4, respectively, Dai et al., reported and demonstrated another smart method called the equivalent PS-FBG [39], which was realized by using a sampled FBG where the phase-shift is equivalently generated by changing the local sampling period instead of the FBG period. The principle of this method is based on the fact that a phase-shift existing in the sampling function can be equivalently regarded as a local phase-shift existing in each of the ghost gratings in terms of the different order of the channel number (i.e., the order of the Fourier series for the sampling function). The superiority of this method over the phase-shift mask one is that only one ordinary phase mask with a sub-micrometer precision instead of the nanometer precision is required in fabrication. However, at the cost of the easy fabrication, the high in-band energy efficiency cannot be obtained by using this method, which means that one should considerably increase the grating length in order to remain a certain reflectivity for the FBG.

### 2.1.2. Applications of the Phase-Shifted FBG to Fiber Lasers

Back in 1994, Kringlebotn et al., first demonstrated an $Er^{+3}$ and $Yb^{+3}$ codoped fiber distributed-feedback (DFB) laser, where a thermally induced PS-FBG was inserted in the cavity and used to narrow the laser emission [22]. Since then, thanks to the unique characteristic of an ultra-narrow band in transmission, the PS-FBGs have widely been used and found applications in fiber lasers, especially in lasers with single longitudinal mode (SLM) emission [22,32,35,36,40–55], where the PS-FBG is generally inserted into the laser cavity and used as an ultra-narrow filter, enabling considerably narrowing the laser emission. To date, various kinds of the PS-FBG-based fiber lasers, such as the distributed Bragg reflector (DBR) lasers [41–43], the distributed-feedback (DFB) lasers [44–47], and the fiber ring lasers [32,35,36,48–55], have been developed. Figure 5 shows several typical examples for the PS-FBGs-based fiber lasers. Figure 5a shows a schematic diagram of a fiber DBR laser reported by Zhao et al. [42], in which the resonant cavity is composed of two identical FBGs (used as the cavity mirror), a PS-FBG, and the $Er^+$ doped fiber. The SLM emission was achieved under the aid of PS-FBG, which was inserted into the FBGs-based laser cavity. Figure 5b shows a typical DFB fiber laser reported by Qi et al. [46], where instead of the two FBGs utilized as cavity mirrors in [42], a $\pi$ phase-shifted asymmetrical FBG was written in an active fiber. Moreover, the light undergoes both the mode coupling and the power gain within the FBG. As a result, the SLM laser emission with a linewidth of 20 kHz and power of 60 mW was obtained, showing that such kinds of lasers have excellent performances, such as a low threshold, high power efficiency, and high optical signal–to–noise ratio, etc.

Compared to the compact DBR and DFB fiber lasers mentioned above, the fiber ring lasers allow more long length and thus more gain in the ring cavity, and as a result, a higher output in laser power can easily be obtained. However, at cost of the high output power, the SLM emission becomes more difficult to be obtained in such kinds of fiber lasers. As an alternative, PS-FBGs have been used and inserted into the fiber ring resonator, which enables considerably narrowing the linewidth of the laser emission. Figure 5c shows a schematic diagram of a PS-FBG-based Erbium-doped fiber (EDF) ring laser [31], where in addition to the PZT-based dynamic PS-FBG, a linearly chirped FBG working in the reflection was used. By suitably controlling the external voltage on PZT, the SLM emission was successfully realized in such kinds of EDF fiber ring lasers.

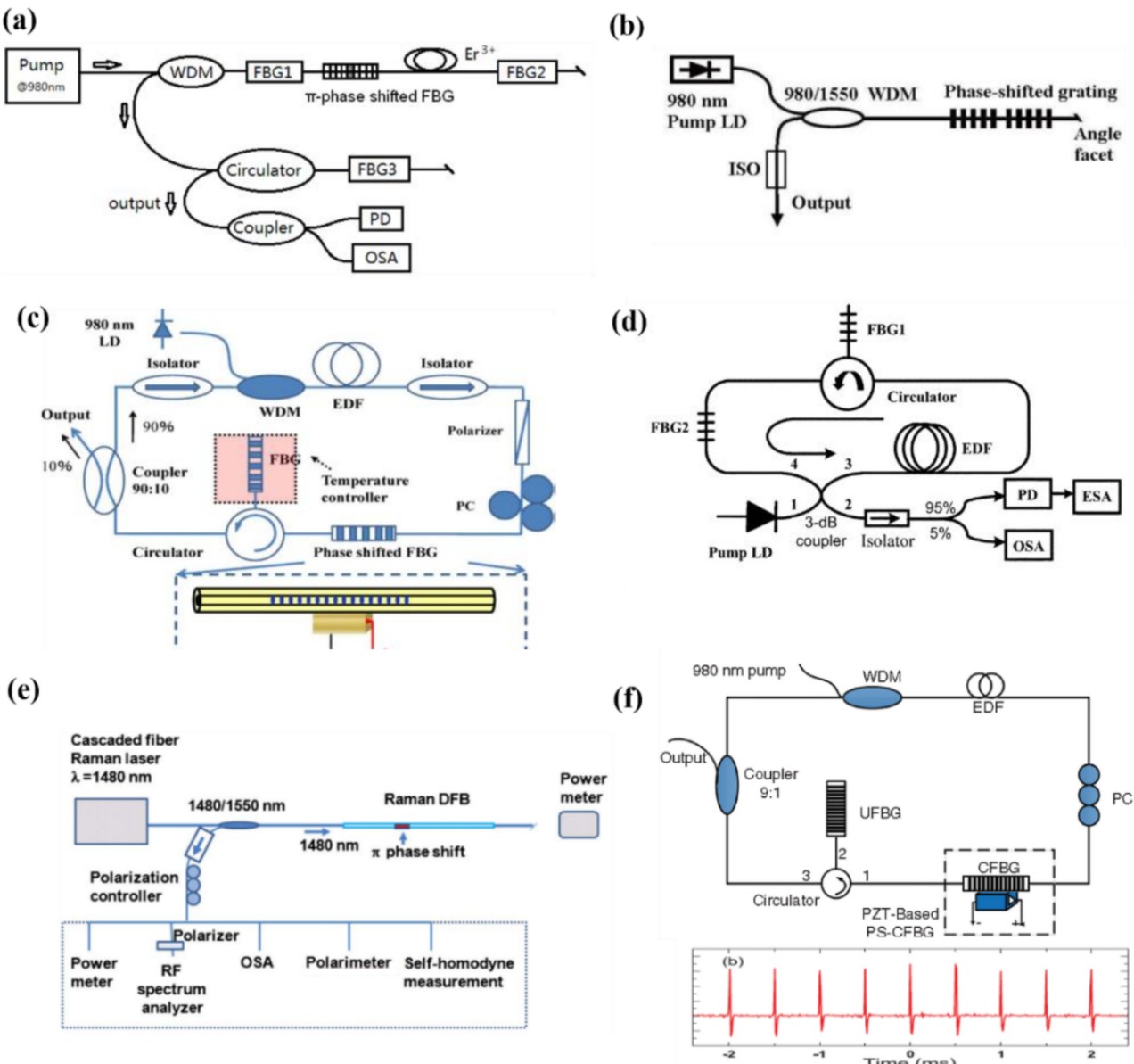

**Figure 5.** (**a**) Schematic diagram of a DFB fiber laser [42]. (**b**) Schematic diagram of an SLM DBR fiber laser [46]. (**c**) Schematic diagram of PZI-induced PS-FBG-based EDF ring laser. Adapted with permission from ref. [31]. (**d**) Equivalent PS-FBG-based EDF ring laser. Adapted with permission from ref. [51]. (**e**) Raman DFBs laser [53]. (**f**) Q-switched EDF ring laser [32].

As a complement to the above fiber lasers, Chen et al. [51] demonstrated another SLM fiber ring laser, as shown in Figure 5d, where the conventional PS-FBG was replaced by an equivalent PS-FBG previously developed by the same authors in [39]. By using the same FBG but with two un-equalized phase-shifts, Sun et al., further demonstrated a dual-wavelength EDF fiber laser [52], where the mode competition between the generated two lasing lines can be well restrained due to the few overlaps of their power spectra.

Besides the above, the PS-FBGs have also been used in other kinds of fiber ring lasers, such as the fiber Raman DFB laser [53], the Q-switched fiber laser [32,54], and the mode-locked fiber laser [55]. Westbrook et al., firstly proposed and demonstrated an FBG-based Raman DFB laser, configuration of which is shown in Figure 5e, where one PS-FBG was used as both the Raman gain medium and the mode-selection filter. Compared with the gain generally produced in an EDF, the intrinsic Raman gain is a distributed one, which has the advantages of wide tunability, being free from the doping constraints, and being

free from the nonlinear-effects; thus, the proposed Raman DFBs would be applied to the fields that cannot be addressed by using conventional DFB fiber lasers [53].

Lastly, Wu et al., have recently proposed and demonstrated a Q-switched EDF ring laser as shown in Figure 5f, where a PZT-based PS-FBG is inserted into the ring cavity and used as an electrical-optical switch. As a typical result, a pulse train with a repetition rate of 1 kHz, pulse width of 15.6 μs, and pulse energy of 1.16 μJ have been obtained under the pump power 90 mW [32].

### 2.1.3. Applications of the PS-FBGs to Microwave Photonics

Thanks to the unique characteristic of an ultra-narrow stop/pass band in reflection/transmission, PS-FBGs have also been found applications in microwave photonics [56–64]. As key components, PS-FBGs have been used to produce either the photonic microwave signals [57–60], the microwave photonic filter [61–63], or the microwave frequency downconverter [64].

In general, a microwave or millimeter-wave signal can be generated in the optical domain based on an optical heterodyning method, in which two optical waves with different wavelengths beat at a photodetector [56]. To obtain a highly stable beat signal, the laser source should be either the single laser but having dual longitudinal modes or the two SLM lasers but operating at two different wavelengths, respectively. In 2006, Chen et al. [57] first proposed and demonstrated a novel approach for the generation of high-frequency microwave signals, shown in Figure 6a, where in combination with a regular FBG, a PS-FBG but with two ultranarrow transmission bands was used to ensure the fiber ring laser operating in SML but with dual-wavelengths. As a typical result, the microwave signal with frequency 18.68 GHz was successfully obtained at the photodetector, as shown in Figure 6b.

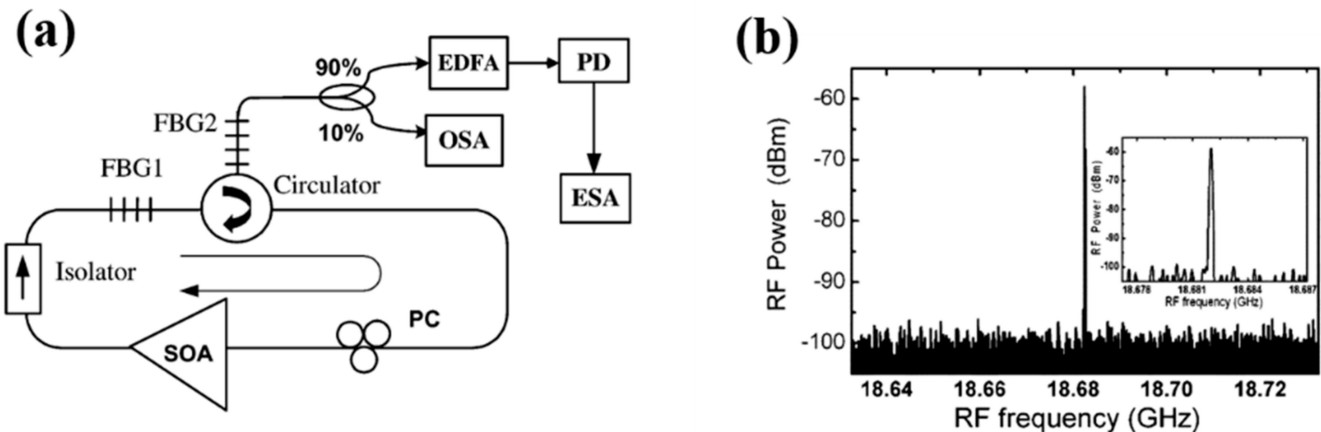

**Figure 6.** (**a**) Schematic diagram of the dual-wavelength SLM fiber ring laser and (**b**) electrical spectra of the generated microwave signal. Adapted with permission from ref. [57]. © [2022] IEEE.

In 2011, Lin et al., proposed and demonstrated another method enabling to generate tunable microwave signals [59], where in combination with a conventional linearly chirped FBG (without the inserted phase-shifts), a linearly chirped FBG, but with two tunable ultranarrow transmission bands, was used to ensure the fiber ring laser operating in SML but with two tunable wavelengths. As a result, the microwave signals with frequencies ranging from 6.88 to 36.64 GHz were successfully demonstrated.

On the other hand, the microwave signals can also be generated by using the external phase/amplitude modulation technique [56]. In 2012, by means of the phase-modulation technique in combination with a PS-FBG, Li et al., proposed and experimentally demonstrated a novel approach that enables the realization of a narrow band and frequency-tunable microwave photonic filter [61], where the PS-FBG was used to convert the phase-modulated signal to an intensity-modulated signal by filtering (removing) one sideband of

the phase-modulated signals. By using the same approach and taking further advantage of two orthogonally polarized optical waves and a PS-FBG, the same group in [61] has demonstrated a new method enabling producing a tunable dual-passband microwave photonic filter (MPF), shown in Figure 7, where the PS-FBG shown in Figure 7a is operated in reflection rather than the transmission. As a result, a dual passband MPF with bandwidth 140 MHz and frequency tunable range of about 6 GHz for both the two passbands was successfully demonstrated, as shown in Figure 7b.

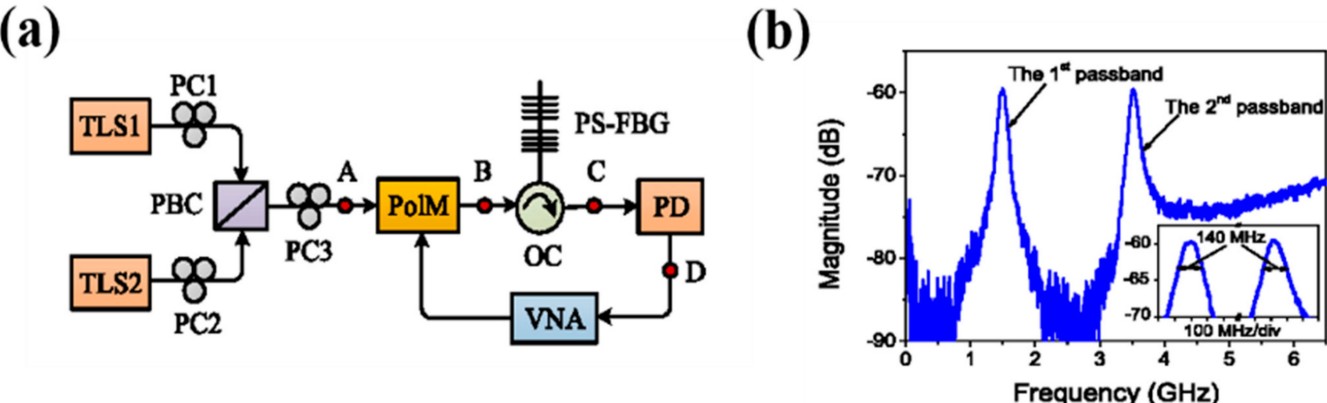

**Figure 7.** (**a**) Schematic diagram of the dual-passband microwave photonic filter and (**b**) spectral response of the microwave photonic filter. Adapted with permission from ref. [63]. © [2022] IEEE.

### 2.1.4. Applications of the PS-FBGs to WDM and Comb Filters

Back in 1994, Agrawal et al. [6] started to investigate the spectral characteristics of the PS-FBG. Based on the result that the peak wavelength of the narrow passband resulted from the PS-FBG is strongly dependent on the amount of phase-shift, they proposed and revealed theoretically that such a kind of FBG can be used as a wavelength de-multiplexer as long as the magnitude of the phase-shift inserted in an FBG can be precisely controlled.

In 2011, the author Li's group experimentally demonstrated a 51-channel comb filter [65], which was realized by using a PZT-induced, phase-shifted multichannel FBG, as shown in Figure 8, where Figure 8a shows the experimental setup for generation of the comb filter, whereas Figure 8b shows the measured transmission spectrum of the comb filter. A multichannel filter with a channel-count up to 51, each with a bandwidth ~27 pm and non-uniformity ~0.3 dB in channel amplitude was successfully obtained. By using the same method, the same authors successfully introduced one, two, and three phase-shifts into a 51-channel, linearly chirped FBG and as a result, three comb filters with channel counts of 51, 102, and 153 were obtained, respectively [66].

Besides the above, the PS-FBG was also designed as a flat-top broadband filter used in transmission, which is strongly desired in applications such as channel selection in a WDM system. Wei et al., firstly proposed and demonstrated such a kind of filter by using a PS-FBG, where the inserted several phase-shifts and their magnitudes were optimally adopted [67].

On the other hand, the parity-time (PT) symmetry Bragg gratings have recently attracted a lot of attention, where in addition to a periodical refractive index-modulation along the grating (i.e., the real part of the index modulation), there also exists a periodical modulation in gain/loss of gating (i.e., the image part of the index modulation) [68]. The results are that some new functions that cannot be provided in conventional Bragg gratings could easily be realized, such as the nonreciprocal filtering, the dispersionless filtering, amplification, and the wavelength demultiplexing as well [68,69]. Most recently, Raja et al., have investigated the spectral features of a phase-shifted PT FBG and demonstrated that such a kind of active FBG could be used as a demultiplexer and mode-selective single-mode laser together in terms of the different working regimes [70].

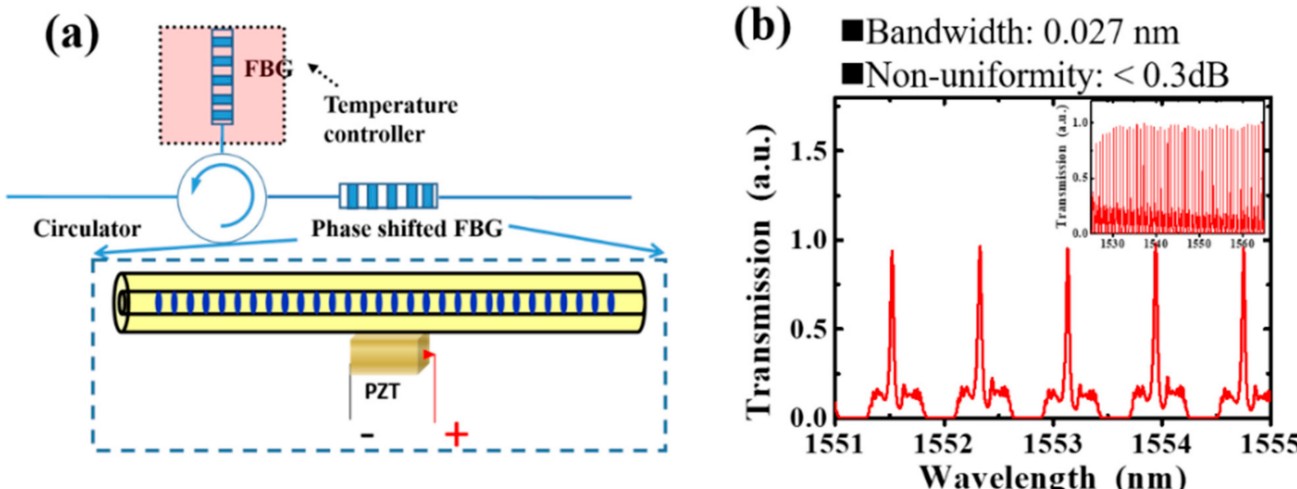

**Figure 8.** (**a**) The experimental configuration for the PS-FBG-based comb filter. (**b**) Transmission spectrum of the comb filter system shown in left side, where the inset shows the whole 51-channel spectrum. Adapted with permission from ref. [65]. © [2022] IEEE.

### 2.1.5. Applications of the PS-FBGs to Optical Switching

The PZT-induced PS-FBG can be used as an electro-optical switch in a Q-switched fiber laser. Compared with the switching based on a uniform FBG, the PS-FBG-based one has a narrower detuning range in wavelength, and thus less switching-time and easier to be implemented in practice. For example, Cheng et al., proposed and demonstrated a Q-switched fiber ring laser, where a PS-FBG combined with an apodized FBG was utilized as a Q-switch [54], i.e., the Q-switched pulses train was obtained by periodically changing the resonance wavelength of the apodized FBG.

Meanwhile, PS-FBGs have also be used as all-optical switching devices [71–77], the principle of which is based on the optical Kerr effect, i.e., the intensity-induced change in the refractive index of the fiber, which could result in an additional phase-shift and thus leads a shift in the peak wavelength as long as the peak power of the incident pulse is strong enough. Back in 1994, Radic et al. [71,72] firstly revealed that an all-optical switching at low intensity can be realized by using a phase-shifted Bragg grating. Lee et al. [73] and Kabakova et al. [74] further analyzed the performance of the PS-FBGs. They both indicated that the PS-FBG has better performance, enabling the higher enhancement in intensity than that of the uniform FBG, and thus allowing the low-power switching. To verify the results above, Melloni et al., experimentally demonstrated a low-power, all-optical switching based on a phase-shifted grating operating at 1.5 μm, an extinction ratio of more than 6 dB was achieved when the peak power of the incident pulse was about 1 kW, which were the first experimental results of the FBG-based all-optical switching [75].

Besides the above works, the theoretical works about the PS-FBG-based all-optical logic gates, e.g., NOT, AND, NAND, OR, XOR, and XNOR gates have recently been completed by Li et al., in [76]. Meanwhile, an optical switch based on graphene-controlled PS-FBG has been reported by Gan et al. [77], where the optical switching with an extinction ratio of 20 dB and a response time of tens milliseconds was successfully achieved.

### 2.1.6. Applications of the PS-FBGs to All-Optical Computing Devices

All-optical computing devices, such as the optical differentiator, optical integrator, and optical Hilbert transformer, etc., are essential to the potential all-optical signal processing and all-optical computer systems [78]. Attributed to the inherent advantages, such as the simplicity, low insertion loss, low-cost, polarization-independence, being free from the electromagnetic interference, and the fully compatibility with the other fiber components, the fiber-based, especially the PS-FBG-based, all-optical computing devices have always been an important topic and attracted increasingly interest in past decades [79–87].

In 2006, Berger et al., for the first time reported and demonstrated an optical temporal differentiator, which was realized by using a π PS-FBG but operating in the reflection [79]. Both the simulation and the experimental results are shown in Figure 9, from which it can be seen that the first-order temporal derivative of an arbitrary optical waveform with a bandwidth of 12 GHz was successfully demonstrated. Almost at the same time, Ngo proposed method to design another π PS-FBG but operating in transmission, which was particularly used for the realization of the first-order optical temporal integrator [80]. By using the same method above, Asghari et al., firstly proposed and numerically demonstrated high-order temporal integrators, which were realized by using multiple-phase-shifted FBGs [81], and the operating bandwidth was estimated to be of tens of GHz.

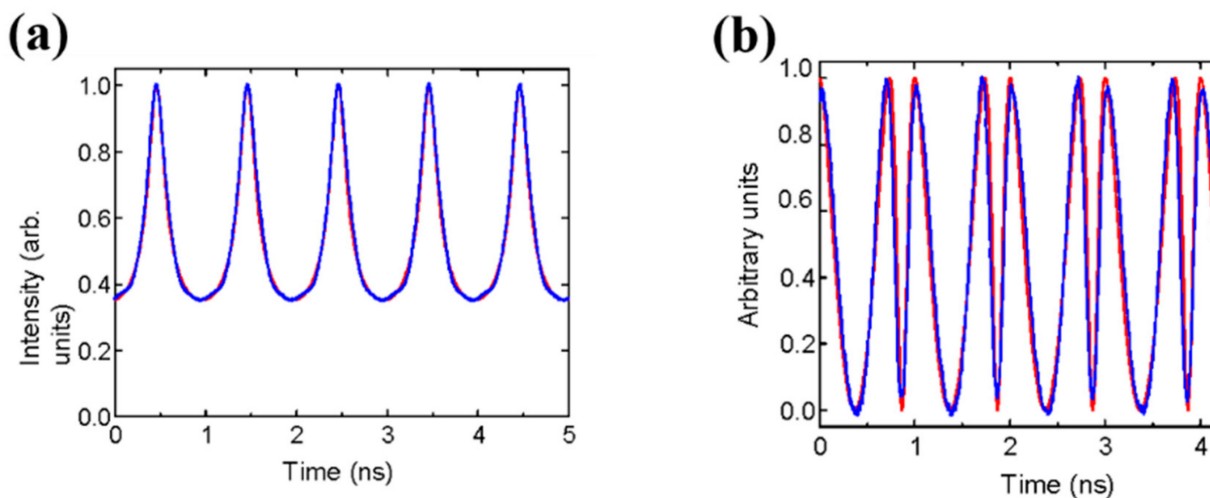

**Figure 9.** Simulation and experimental results for the PS-FBG-based differentiator. (**a**) Intensity profiles of (**a**) input pulses and (**b**) output pulses. Adapted with permission from ref. [79]. © The Optical Society.

As one of the important all-optical computing elements, Hilbert transforms have also been realized in the optics domain by using FBG-based optical devices. In 2009, Asghari et al., firstly proposed and numerically demonstrated a fiber-based Hilbert transformer [83], where an apodized-shifted FBG was utilized. In 2010, Li. et al. [84] proposed an all-fiber temporal fractional Hilbert transformer (FHT), where a multiple-phase-shift FBG but with an addition of Sinc-like apodization was utilized. As a result, an all-optical FHT with a bandwidth of up to hundreds of gigahertz was successfully obtained, no matter how complex the initial optical waveform would be.

Most recently, Liu et al., have proposed and demonstrated a new kind PS-FBG that can be simultaneously used as a temporal first-order optical differentiator and a temporal first-order optical integrator [87], i.e., the proposed FBG is served as a differentiator in its reflection, meanwhile in the transmission, the grating is served as an integrator. Simulation results for both of these functions are shown in Figure 10.

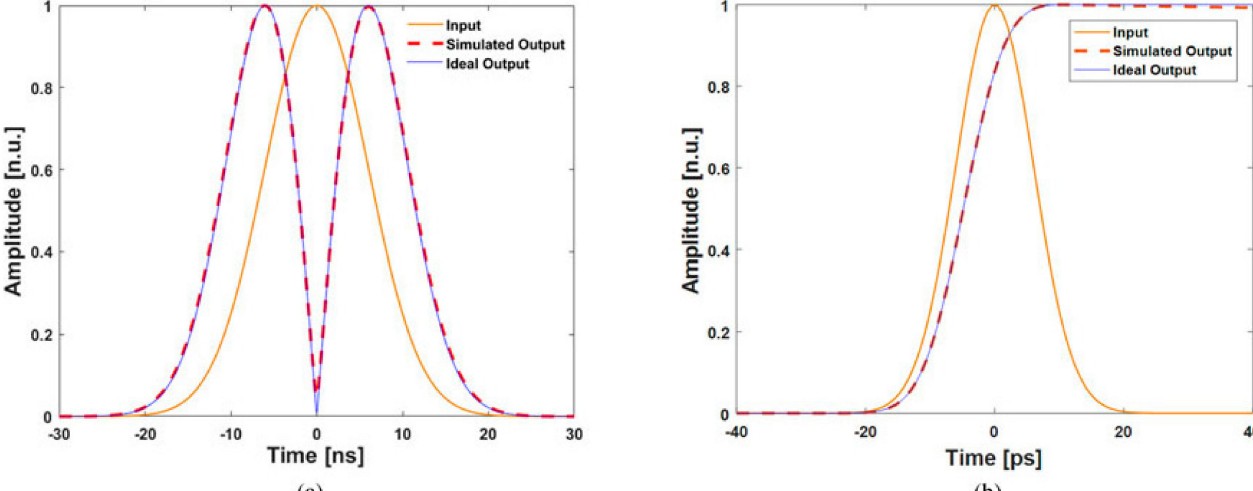

**Figure 10.** Simulation results for both the differentiation and the integration. (**a**) Input and output of the differentiator. (**b**) Input and output of the integrator. Adapted with permission from ref. [87]. © [2022] IEEE.

### 2.1.7. Applications of the PS-FBGs to Fiber Sensors

For a PS-FBG, there always exists an ultra-narrow notch in its reflection band. Characteristics of such a notch including the wavelength of the loss-peak and the notch depth have nearly a linear response to the magnitude of the induced phase-shift [49], which makes the PS-FBG an excellent candidate of either the wavelength- or the power-interrogated sensor, enabling providing more versatile and higher accuracy measurements than those of the conventional FBGs. To date, various kinds of the PS-FBG-based sensors have been proposed and demonstrated, which for examples include the torsion sensors [17], transverse load sensors [88–90], strain sensors [91,92], bending sensors [93], pressure sensors [94,95], temperature sensors [21,49,95], refractive index sensors [37,38,95], acoustic sensors [96–98], and magnetic field sensors [99], etc. The sensing principle underpinning these sensors relies on the fact that changes in the environmental parameters, e.g., the applied torsion, temperature, and the ambient refractive index, etc., would result in change in the inserted phase-shift of the FBG, which thus lead the change in depth and shift in peak wavelength of the notch. Back in 1999, by using a π PS-FBG, LeBlanc et al., firstly proposed and demonstrated a temperature-insensitive transverse-load sensor, shown in Figure 11a [88], where the load applied in direction transverse to the fiber axis can be determined with a high accuracy by measuring the birefringence-induced spectral splitting of the ultra-narrow passband of the PS-FBG itself, as shown in Figure 11b,c, respectively. As typical result, a distributed force with a resolution of $1.4 \times 10^{-3}$ N/mm was successfully measured, which corresponds to a strain-change of 0.5 με in fiber core. To further increase the sensitivity of the PS-FBG-based transverse-load sensor, in 2009, Fu et al., proposed and demonstrated an alternative approach [89], where the photonic microwave signal originated from the PS-FBG-based dual-wavelength laser (i.e., the beat signal of the two-wavelengths laser) was utilized to measure the load-induced change in the frequency of the microwave. Furthermore, in 2015, Wang et al., proposed and demonstrated another improved method [90], where instead of the amplitude-spectrum measurement in [89], a measurement on the polarization-dependence-loss (PDL) spectrum of the PS-FBG was utilized; as a result, a higher load sensitivity than the previous one in [89] was successfully obtained.

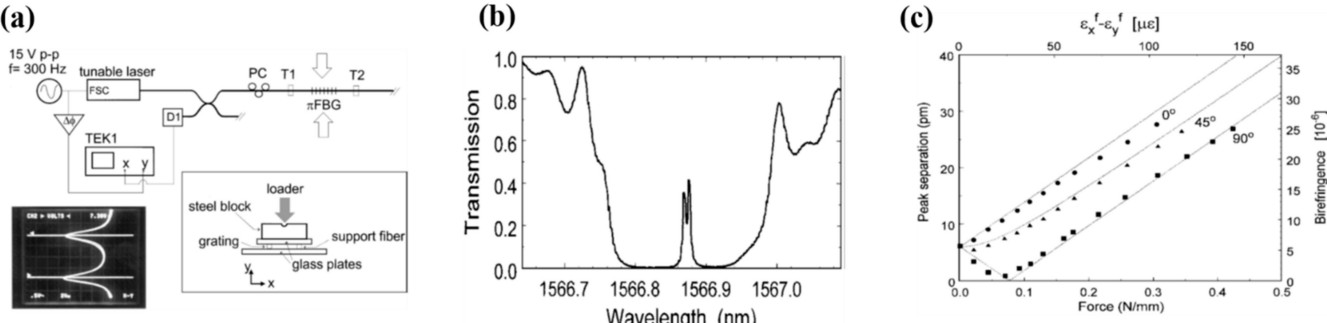

**Figure 11.** (**a**) The proposed PS-FBG-based sensing system for measurement of the transverse load along a fiber. (**b**) Transmission spectra of the adopted PS-FBG without load. (**c**) The relationship between the peak wavelength separation (corresponding to the birefringence) and the applied forces in transverse direction. Adapted with permission from ref. [88]. © The Optical Society.

In 2016, Huang et al. [17] proposed and experimentally demonstrated a temperature- and strain-independent torsion sensor, which was realized by using a PS-FBG inscribed in a conventional fiber with the line-by-line femtosecond-lase technique. Due to the strong birefringence of the grating induced during the side-writing process, there exists two peaks in the reflection band, i.e., $P_1$ and $P_2$ as shown in Figure 12a. The authors revealed that the difference between the two transmission peaks ($P_1 - P_2$) has a sinusoid function-like relationship with the torsion applied on the PS-FBG as shown in Figure 12b. As a typical result, a PS-FBG-based torsion sensor with sensitivities of 963.53 dB/(rad/mm) and $-1032.71$ dB/(rad/mm) for clockwise and count-clockwise directions, respectively, was successfully demonstrated.

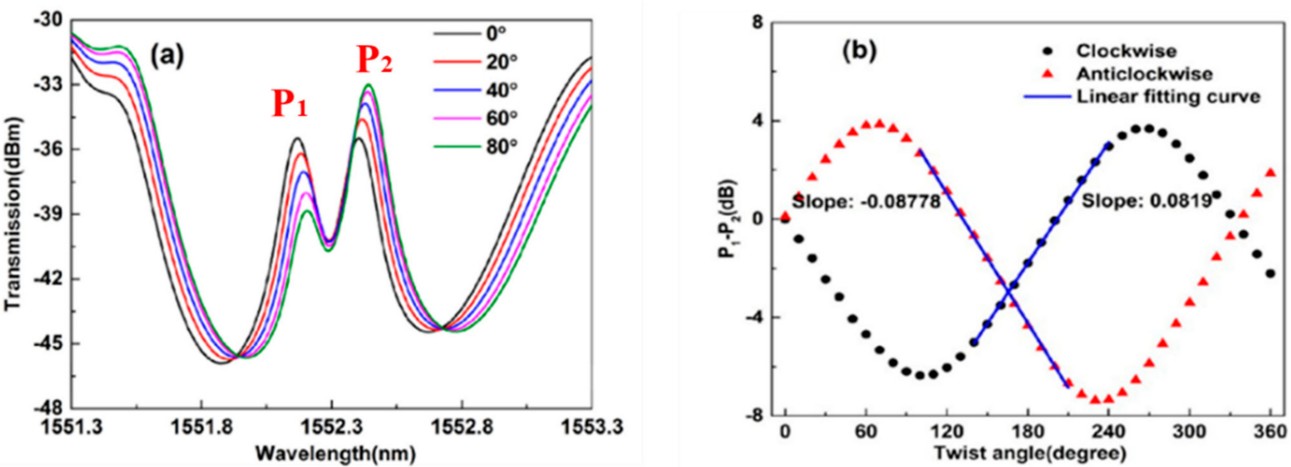

**Figure 12.** (**a**) Transmission spectra of the proposed PS-FBG under different torsion angles. (**b**) The relationship between the transmission difference $P_1 - P_2$ and the applied torsion. Adapted with permission from ref. [17]. © The Optical Society.

In combination with other sensing elements, the PS-FBG can also be used as a multi-function sensor, enabling measuring multiple parameters simultaneously. In 2020, Yang et al., proposed and experimentally demonstrated a novel sensor enabling the simultaneous measurement of the refractive index, temperature, and pressure, which is based on a hybrid structure consisting of a Fabry–Perot interferometer (FPI) and a PS-FBG as shown in Figure 13 [95], where the FPI was formed by creating a microcavity at middle of the FBG, as shown in Figure 13a. By measuring the three parameters, the wavelength change in one of the FPI peaks and the changes in both the notch depth and notch wavelength of the PS-FBG, as shown in the inset of Figure 13b, the ambient temperature, pressure, and the refractive

index of the liquid inside the EPI can be indirectly determined while the crosstalk effects among these three-parameters measurements can be completely eliminated. As a typical result, a multiple-parameter sensor with a temperature sensitivity of 307.6 pm/°C, pressure sensitivity of 0.81 nm/MPa, and RI sensitivity of 355.03 nm/RIU has been successfully demonstrated.

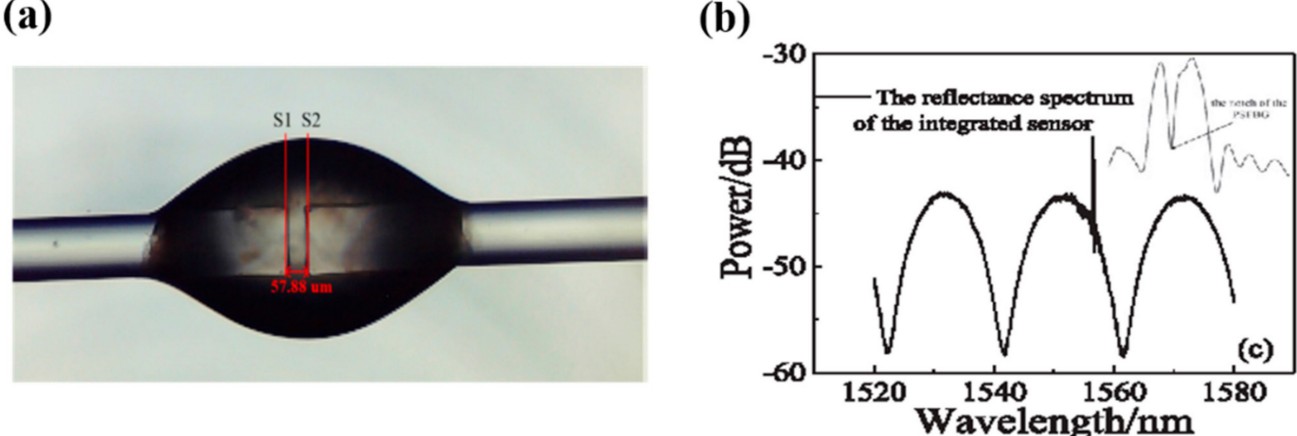

**Figure 13.** (**a**) Schematic diagram and (**b**) reflectance spectrum of the FBG-FP-based sensor for simultaneous measurement of three-parameter. Adapted with permission from ref. [95].

In addition to the sensing applications above, the PS-FBGs have also been explored to find applications in the measurement of the ultrasonic and magnetic field, etc. In 2011, Rosenthal et al., proposed and demonstrated a highly sensitive, yet compact, hydrophone, shown in Figure 14a, where a PS-FBG was used for measurement of a wideband ultrasonic field. The principle of this method lies in the fact that the measured ultrasonic filed could be equivalently expressed as a transverse pressure applied on the FBG, whereas the peak wavelength of the narrow notch resulted from the PS-FBG is strongly dependent on the transverse-load pressure [88–90]; thus, a change in such a peak wavelength can be used to measure ultrasound-induced pressure variations within the grating. Some of the measurement results are shown in Figure 14b. In contrast to standard fiber sensors, the high finesse of the notch, which is the reason for the sensor's high sensitivity, is not associated with a long propagation length. As a typical result, a PS-FBG-based ultrasonic sensor with an effective length of 270 μm, a pressure sensitivity of 440 Pa, and an effective bandwidth of 10 MHz was successfully demonstrated. To further improve the stability and sensitivity of the ultrasonic sensing system, in 2015, Guo et al., proposed and demonstrated another PS-FBG-based ultrasonic sensor [98], where the sensing system was more robust and more stable since the wavelength (frequency) of the utilized laser was precisely locked to the notch wavelength of the utilized PS-FBG by using the Pound–Drever–Hall (PDH) technique. The proposed method enables considerably reducing the effect of laser intensity noise and thus enables high-sensitivity ultrasonic detection.

The PS-FBGs have also been used to measure the magnetic field. In 2018, Bao et al., proposed and demonstrated a PS-FBG-based magnetic field sensor [99], where a magnetic fluid (MF) was infiltrated into gap of the FBG as shown in Figure 15a. The principle of this proposed method relies on the fact that the change in the external magnetic field will result in a linear change in the refractive index (RI) of the MF, and thus bring changes in the notch wavelength and the depth of the PS-FBG, which in return means that changes in peak wavelength and depth of the notch can be used to measure the external magnetic field. Some of their experimental results are shown in Figure 15b; it can be seen that the magnetic field sensor with a sensitivity of 2.42 pm/Oe at an amplitude ranging from 0 to 120 Oe has been successfully demonstrated.

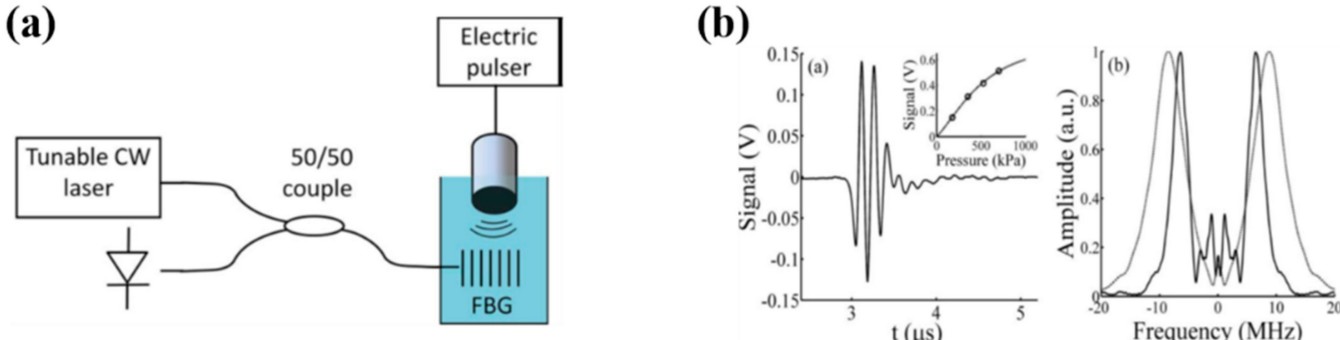

**Figure 14.** (**a**) Schematic description of the ultrasonic field measurement. (**b**) Temporal and spectral responses of the optical sensor obtained for an acoustic plane wave formed by an ultrasound transducer. Adapted with permission from ref. [96]. © The Optical Society.

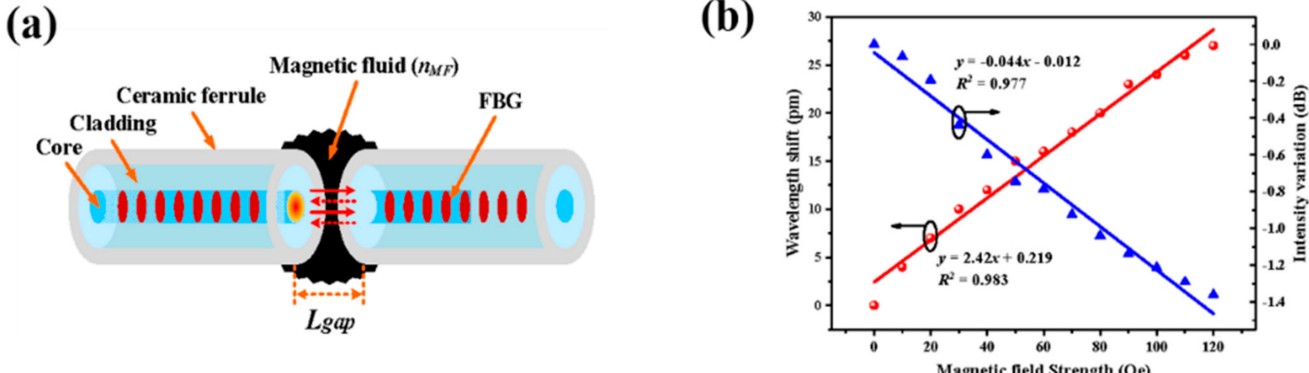

**Figure 15.** (**a**) Schematic diagram of the proposed magnetic field sensor. (**b**) Wavelength shift and intensity variation of the transmission peak against magnetic field strength. Adapted with permission from ref. [99]. © [2022] IEEE.

In addition to the above applications, the authors' group proposed a method enabling quantitatively calibrating a phase-shift formed in a linearly chirped FBG [49]. A new approach for temperature measurement based on the proposed PS-FBG was experimentally demonstrated accordingly. Referring to the PS-FBG-based temperature sensors, Hnatovsky et al., have recently proposed and demonstrated another PS-FBG named the π-shifted Type II FBG [21], which is written in a conventional SMF fiber using the infrared femtosecond laser. The proposed π-shifted type II FBG exhibits a remarkable thermal stability at a temperature even up to 1000 °C. Such a kind of PS-FBGs is expected to find applications to high-resolution measurements of temperature, load, and vibration under extreme high-temperature environments.

Last but not least, in recent years, the fiber lasers operating at a wavelength around 2 μm have attracted extensive attention and have been found applications beyond fiber communications, such as in the military, medicine, optical sensing, and optical measurement due to the superior properties of such lasers, e.g., safe for the eyes and strong absorption for various gas molecules [100]. Zhang et al., have recently investigated the transmission characteristics of PS-FBG and chirped PS-FBG working at a wavelength around 2 μm both theoretically and experimentally; of the results, particularly, three narrow peaks with a narrow bandwidth (less than 0.09 nm) have been successfully demonstrated [101,102].

### 2.2. Phase-Only Sampled FBGs and Their Applications

The original idea of the sampled grating was proposed and demonstrated by Jayaraman et al., in 1993 for producing a multi-wavelength, semi-conductor laser [103]. Ouellette et al., firstly exploited such a sampling method to a linearly chirped FBG in order to realize

multichannel dispersion compensation [104], where the most straightforward sampling function, i.e., a periodic rectangular function, was utilized. In a spectral domain, the rectangular sampling function corresponds to a sinc envelope, which modulates the amplitude of the multiple channels, resulting in a high nonuniformity for the resulted multiple channels. The sinc-type sampling [105] can overcome such nonuniformity problems; however, fabrication of the sinc-sampled FBG requires a highly precise control of both the amplitude and the phase of the grating, which is extremely difficult to be realized in fabrication, especially when the channel count required is larger than 16.

Phase-only sampling can help us to fully overcome such issues existing in the amplitude sampling [106–119]. The phase-only sampled FBG was proposed and firstly demonstrated by Rothenberg and Li et al., in [106,107], where a linearly chirped FBG with a channel count up nine was firstly demonstrated both theoretically and experimentally. Compared with the sampling method in ref. [105], where extremely high index modulation and a complex envelope (apodization) in the amplitude of the seed grating itself are desired, the phase-only sampled FBG has no modulation in its amplitude such that the apodization profile for a multichannel FBG is the same as that of the seed grating; meanwhile, the maximum index modulation required for a 45- or 81-channel FBG can be reduced to a really accepted level [114], which thus makes the phase-only sampled FBG particularly suitable to be fabricated with the robust side-writing phase mask technique. To date, the phase-only sampling method has become an established yet the unique technique that allows the practically fabrication of the multichannel FBG with a channel count up to 81 [114].

### 2.2.1. Principle and Optimization of the Phase-Only Sampled FBG

Phase-only sampled FBG refers to an FBG whose amplitude (i.e., the maximum index modulation of the grating) is modulated by a slowly changed phase-only function $s(z)$. In combination with Equation (1), the index modulation distribution for a phase-only sampled FBG $\Delta n_M(z)$ can be expressed as

$$\Delta n_M(z) = \text{Re}\left\{ \frac{\Delta n_1(z)}{2} \exp\left( i\frac{2\pi}{\Lambda_0}z + i\phi_g(z) \right) \cdot s(z) \right\} \tag{3}$$

Since $s(z)$ is a periodic phase-only function, it can be further expanded in the Fourier series as

$$s(z) = \exp[i\theta(z)] = \sum_{m=-\infty}^{\infty} S_m \exp\left( i\frac{2m\pi}{P}z \right) \tag{4}$$

where $\theta(z)$ and $P$ represent the phase and sampling period of the function $s(z)$, respectively. $S_m$ represents the $m$th Fourier coefficient. By substituting Equation (4) into Equation (3), one can obtain

$$\Delta n_M(z) = \sum_{m=-\infty}^{\infty} |S_m| \text{Re}\left\{ \frac{\Delta n_1(z)}{2} \exp\left( i2\pi\left( \frac{1}{\Lambda_0} + \frac{m}{P} \right)z + i\phi_g(z) + i\varphi_m \right) \right\} \tag{5}$$

where $\varphi_m$ is the phase of the coefficient $S_m$. To take into careful account Equation (5), one can easily find that $\Delta n_M(z)$, i.e., the phase-only sampled HLPG, can be equivalently regarded as a linear superposition of an infinite number of the gratings (channels), and each of these gratings has its own amplitude of $|S_m|\Delta n_1(z)/2$ and its own period of $\Lambda = 1/(1/\Lambda_0 + m/P)$ in terms of different orders of the Fourier series $m$. The channel spacing (in unit of the wave number) among these individual gratings remains a constant of $1/P$ (corresponding to a wavelength spacing of $\lambda_0(\Lambda_0/P)$, where $\Lambda_0$ and $\lambda_0$ are the central period and resonant wavelength of the seed FBG, respectively). Therefore, as long as the channel spacing adopted is larger than the bandwidth of the seed FBG itself, multiple channels without any inter-channel interferences could be obtained by using the sampling method. In most of the application cases, especially for the multichannel FBG used in WDM system, high channel uniformities (including the uniformities in channel bandwidth, channel amplitude, and channel spacing) and high channel count are strictly required,

which in turn means that, for a phase-only sampled FBG with a required channel of $2M + 1$, the sampling function $s(z)$ needs to be optimized such that all the in-band Fourier coefficients $|S_m|$ remain a constant and should be close to $1/\sqrt{2M+1}$ as possible [107]. In general, the cost function enabling satisfying the above design criterions for the function $s(z)$ shown in Equation (4) can be expressed by [114]

$$E(z) = \sum_{m=-M}^{M} \left[ |S_m|^2 - \frac{\eta}{2M+1} \right]^2 \tag{6}$$

where $\eta$ is the target diffraction efficiency for all the in-band channels of $(2M + 1)$. $S_m$ are the Fourier series of the sampling function $s(z)$. The summation $\sum_{m=-M}^{M} |S_m|^2$ is generally called the diffraction efficiency, which actually represents the in-band energy efficiency of the phase-only sampling function.

In 2003, the author Li et al., firstly proposed and systematically demonstrated the phase-only sampling method to produce a multichannel FBG [107], where a discrete phase-only function including either the binary-level one (also called the Dammann phase-only sampling function) or the multi-level one was particularly selected and optimized by using the simulated annealing algorithm. Figure 16 shows the design results for a 39-channel FBG, where the Fourier spectrum of the optimized phase-only function and the calculated reflection spectrum of the corresponding 39-channel FBG are shown in Figure 16a,b, respectively. From these two figures, it can be seen that the channel uniformities are strongly related to the Fourier transform of the phase-only sampling function. The channel nonuniformity is less than 2%, and in-band energy efficiency is about 80%, which in turn means that nearly 20% of the out-of-band channels are inevitably generated; thus, such a kind of multichannel FBG can be used only to those applications where the out-of-band channels can be allowed to exist.

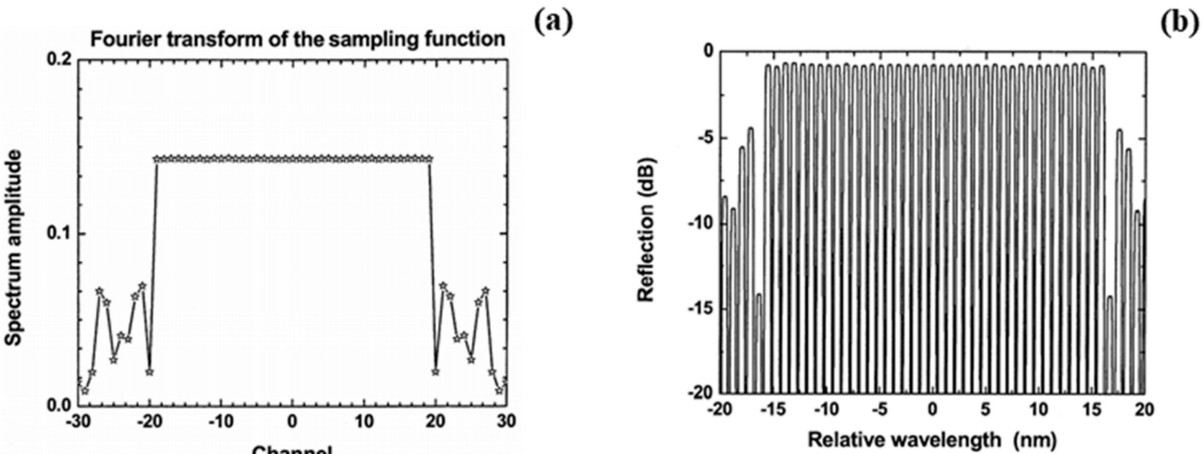

**Figure 16.** Design results for a binary phase-only sampled FBG. (**a**) Fourier spectrum of the binary phase-only sampling function (one sampling period). (**b**) Reflection spectrum of the 39-channel FBG. Adapted with permission from ref. [107]. © [2022] IEEE.

To decrease those unnecessary out-of-band channels resulting from the discrete phase-only sampling, the author Li et al., further proposed and demonstrated a continuous phase-only sampling method [107], where the phase-distribution $\theta(z)$ of the sampling function in Equation (4) was assumed to have the form including a certain number of harmonic terms expressed by [107]:

$$\theta(z) = \sum_{n=1}^{J} \alpha_n \cos(2\pi nz/P) \tag{7}$$

where $\alpha_n$ are the free parameters that need to be optimally selected such that the cost function in Equation (6) becomes the minimum, i.e., each of the Fourier coefficients $S_m$ in Equation (5) are identical within the considered in-band channels. Based on the symmetric characteristics of the Fourier series $S_m$ in Equation (4), it can be mathematically proven that the in-band identical channels up to the $2J - 1$ can be expected to achieve once if the $J$ harmonic terms are adopted in Equation (7). Based on the continuous phase-only sampling method, a 45-channel FBG with an excellent channel uniformity (99.2%) and high in-band energy efficiency (92%) was numerically demonstrated [111]; the results are shown in Figure 17, where the phase distribution and Fourier spectrum of the phase-only sampling function are shown in Figure 17a,b, respectively.

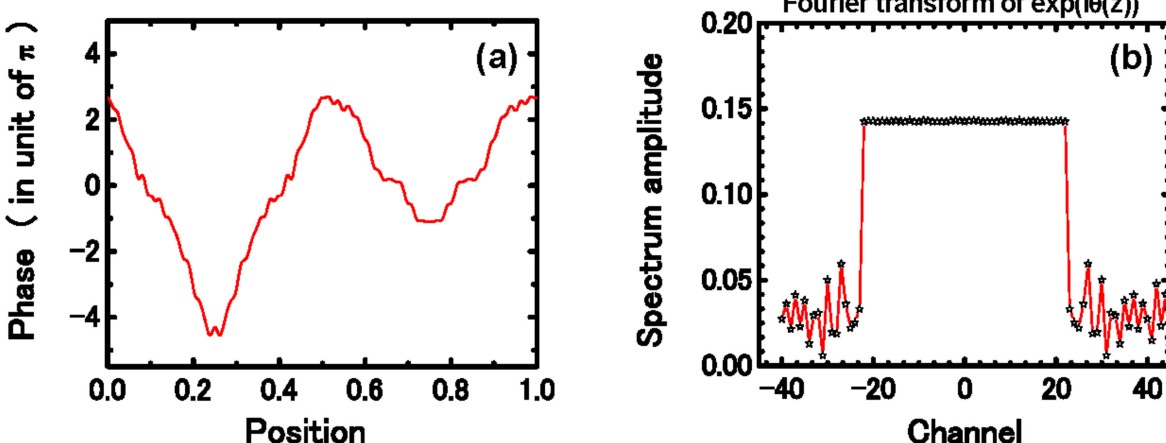

**Figure 17.** Design results for a continuous phase-only sampling function. (**a**) Phase distribution of the optimized 45-channel phase-only sampling function. (**b**) Fourier spectrum of the corresponding phase-only function. Adapted with permission from ref. [111]. © The Optical Society.

To further increase the channel numbers enabling covering the whole S, C, and L bands subsequently, Li et al., proposed and demonstrated a nonconsecutive 135-channel FBG [116], which was realized by simultaneously utilizing two phase-only sampling functions, namely the double phase-only sampling method; the principle schematic is shown in Figure 18, where $S_1$ and $S_2$ represent the first and the second phase-only sampling function, respectively, which have channel counts of M and N, respectively. The total channel count of the double-sampling function then becomes M × N as long as the condition that the sampling period $P_1$ (i.e., period of the function $S_1$) is M-fold of $P_2$ (i.e., period of $S_2$) is satisfied. Figure 19 shows the calculated results for a doubly sampled 135-channel FBG, where Figure 19a shows the reflection spectrum for all 135 channels, and Figure 19b shows the reflection and group delay spectra of the central three channels within the central C band. It can be seen that a multichannel FBG with a channel count of 135 and channel spacing of 0.8 nm has been numerically demonstrated. Each of the channels has an identical strength of 10 dB (90% reflection) and an identical bandwidth of 0.4 nm @ 0.5 dB through all the 135 channels, which enables covering the whole S, C, and L bands of fiber communication.

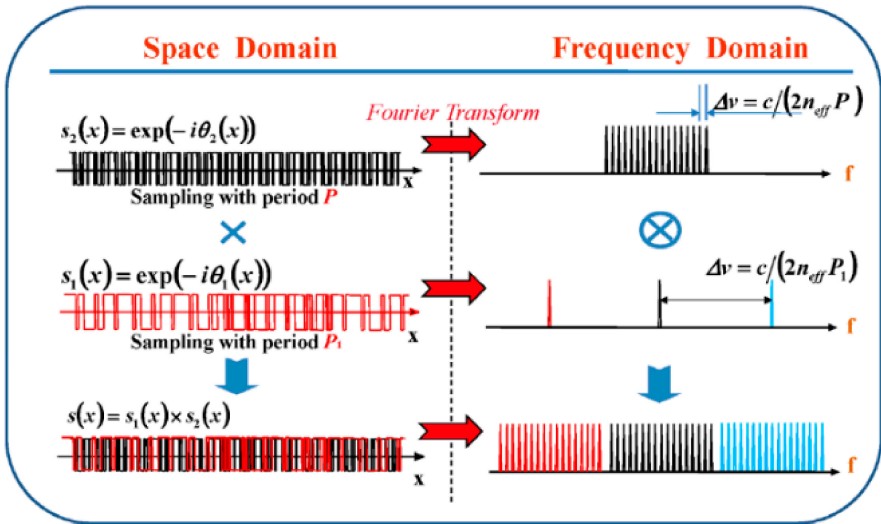

**Figure 18.** Principle of the double phase-only sampling method based on the Fourier analysis. Adapted with permission from ref. [116]. © The Optical Society.

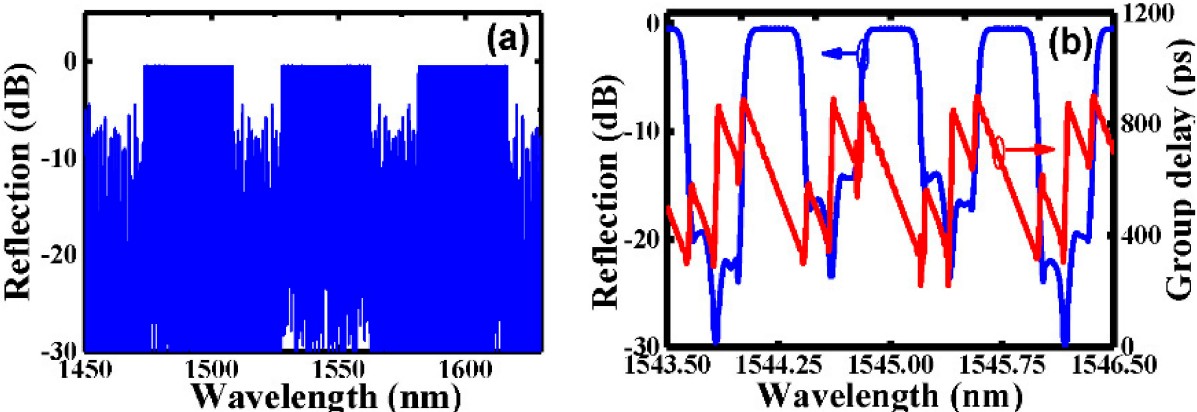

**Figure 19.** Design and the calculated results for the sampled 135-channel FBG. (**a**) Reflection spectrum for all 135 channels and (**b**) reflection and group delay spectra within the central three channels of the central band in Figure (**a**). Adapted with permission from ref. [116]. © The Optical Society.

2.2.2. Fabrication and Experimental Results for the Designed Multi-Channel FBG

Unlike the PS-FBG, where few number of the phase-shifts are needed to insert into the FBG, in the phased-only sampled FBG, a huge number of the phase-shifts are required to insert into the seed FBG; therefore, the phase mask-based proximity side-writing technique is generally utilized in the fabrication process [107], where the phase of the sampling function $s(z)$ is especially encoded into a conventional phase mask, namely the phase-shifted phase mask. Concretely, once the phase of the sampling function is optimally determined, it can be discretely encoded into each local period $\Lambda_M(z)$ of the phase-shifted phase mask (phase grating) by using the following equation [107,114]:

$$\Lambda_M(z) \approx \frac{2}{\Lambda_0^{-1} + (1/2\pi)(d\Phi(z)/dz)} \tag{8}$$

where $\Lambda_0$ is period of the seed FBG, and $\Phi(z) = \theta(z) + \phi_g(z)$ represents summation of the two phases, i.e., phase of the sampling function $\theta(z)$ and the local phase of the seed grating $\phi_g(z)$. Basically, there are two approaches enabling implementing the $\Lambda_M(z)$. Ideally, one can change every period of the grating according to Equation (8). In this case, each period of the phase grating must be continuously written in the phase mask without stitching

error, and thus high-accuracy etching techniques such as the e-beam or laser-beam etching ones are essential to fabricating such a complex phase mask. As an alternative, one can also divide the entire grating into a large number of steps. Each step is considered a uniform grating with the local average period $\Lambda_M(z)$. The latter approach is an extension of the conventional step-chirped grating to the sampled and chirped grating, which is easier to be realized; however, the stitching errors between each of the two neighbor steps cannot be avoided, which would cause large ripples in both the reflection and group-delay spectra. More importantly, some studies have shown that when the phase-shifted phase mask is used to write multichannel FBG, the phase-shift in mask is not directly replicated into the FBG [120]. Based on a rigorous finite difference in time domain, Sheng et al., analyzed the near-field diffraction of a phase-shifted phase mask [121] and proposed a pre-compensated method for binary phase-only sampling. Soon after, Li et al., proposed and demonstrated a pre-compensated phase-shifted phase mask for the successful fabrication of a high-channel-count FBG with channels up to 81 [112].

By using the former approach where the phase mask was designed and fabricated with a special "stitch-error-free" lithography tool [107,112], the author Li's group successfully fabricated and demonstrated 45- and 81-channel FBGs; the measurement results are shown in Figure 20, and have been the unique yet best results among those from all the other multichannel FBGs reported to date.

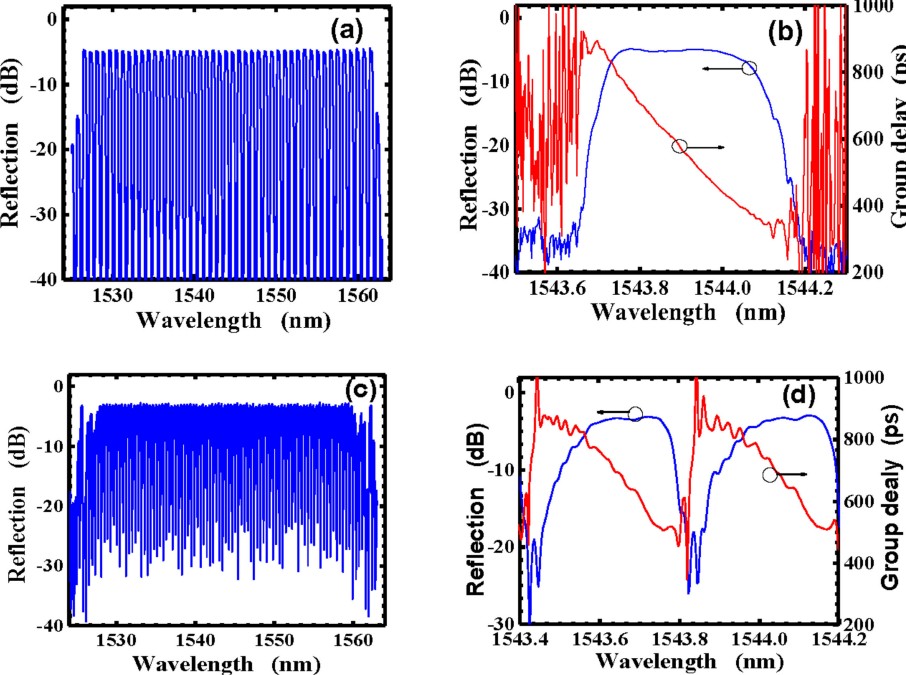

**Figure 20.** Measurement results for 45- and 81-channel FBGs. (**a**) Reflection spectrum for the 45-channel FBG and (**b**) reflection and group delay spectra in central channel of the 45-channel FBG. (**c**) Reflection spectrum for the 81-channel FBG and (**d**) reflection and group delay spectra in the central channel of the 81-channel FBG. Adapted with permission from ref. [112]. © The Optical Society.

### 2.2.3. Applications of the Phase-Only Sampled FBG in the Fields of Optical Communications and Optical Signal Processing

Attributed to characteristics of the high channel count and the high uniformities in both intra- and the inter-channels, the phase-only sampled FBGs have found various applications in the fields of optical communication and optical signal processing. To date, the phase-only sampled FBGs have been used as comb filters [65,122], broadband chromatic dispersion compensators [107,108,111,113,114,123], WDM interleavers [124], and multichannel differentiators [125], etc.

In 2009, the Li et al., first proposed and demonstrated a comb filter with a channel count of up to 153 [65,66], which was realized by using a phase-only sampled FBG (operating in reflection) and a π phase-shift phase-only sampled FBG but operating in its transmission. By inserting such a kind of comb filter into the fiber ring amplifier, the Li et al., further proposed and demonstrated a fiber laser, enabling the multiwavelength emission; output spectra of the proposed laser are shown in Figure 21 [122]. From this, it can be been that a fiber laser with more than 30 wavelengths lasing was successfully demonstrated, whereas the 3 dB linewidth for each of the lasing lines is about 13 pm and the signal–to–noise ratio is larger than 50 dB.

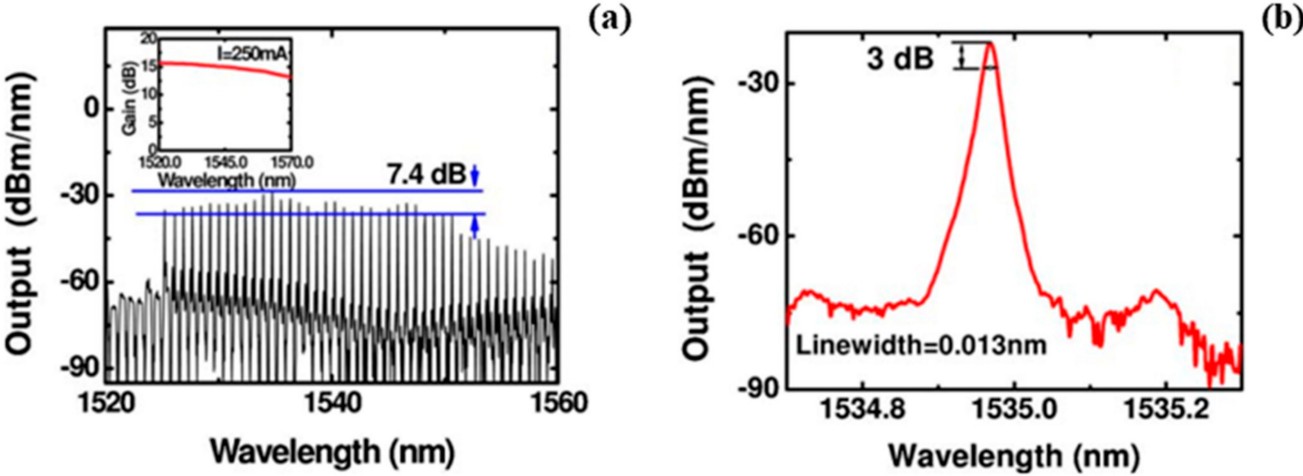

**Figure 21.** Output spectra of the comb filter-based multiwavelength fiber laser. (**a**) Laser emissions with 51 channels and (**b**) emission in one of the channels. Adapted with permission from ref. [122]. © The Optical Society.

With the increasing demands of the wider band and the higher date-rate signals in long-haul fiber link system, the synchronous compensations for fiber dispersion have become the inevitable issue to be overcome. In the early times of 1990, the chirped FBGs were original proposed and used to compensate the fiber dispersions, which mainly can be classified into two categories, i.e., the linearly chirped and the nonlinearly chirped FBGs. Of these, the former one can only be used as a dispersion compensator, whereas the latter one can be synchronously used as the dispersion and the dispersion-slope compensator, but both with a narrow band [123]. In 2003, by applying the phase-only sampling method to a one-channel linearly chirped FBG [107], Li et al., firstly proposed and experimentally demonstrated a five-channel dispersion and dispersion-slope compensator, some of the results are shown in Figure 22, from which it can be seen that in each of the fiber channels, the dispersion of 1020 ps/nm and dispersion slope 5000 ps/nm$^2$ were successfully obtained, which can exactly be used to compensate the dispersion of 60 km SMF fiber.

To further increase bandwidth of the FBG-based dispersion and dispersion-slope compensator, several years later, the Li's group proposed and demonstrated a new kind of phase-only sampled FBG with channels up to 51, where both the grating period and the sampling period were changed (chirped) linearly along the length of grating, as a result enabling the simultaneous dispersion and dispersion-slope compensator to cover the whole C-band as much as it could be expected to obtain. Moreover, the optimized sampling function was a continuous one with an analytical form; thus, the split effect of phase-shifts in the phase mask caused by the diffraction [120,121] can easily be compensated. The design results are shown in Figure 23 [115], where the dispersion and the dispersion slope at wavelength 1545 nm are about the 1815 ps/nm and 6.6 ps/nm$^2$, respectively, which in turn means that the proposed device enables the dispersion compensation of the 110 km SMF fiber coving the full C-band. Lee et al., also revealed and numerically demonstrated that in addition to the broadband dispersion compensation function, the phase-only sampled

FBG could also be used to design multichannel add-and-drop and channel interleavers in a WDM system [124].

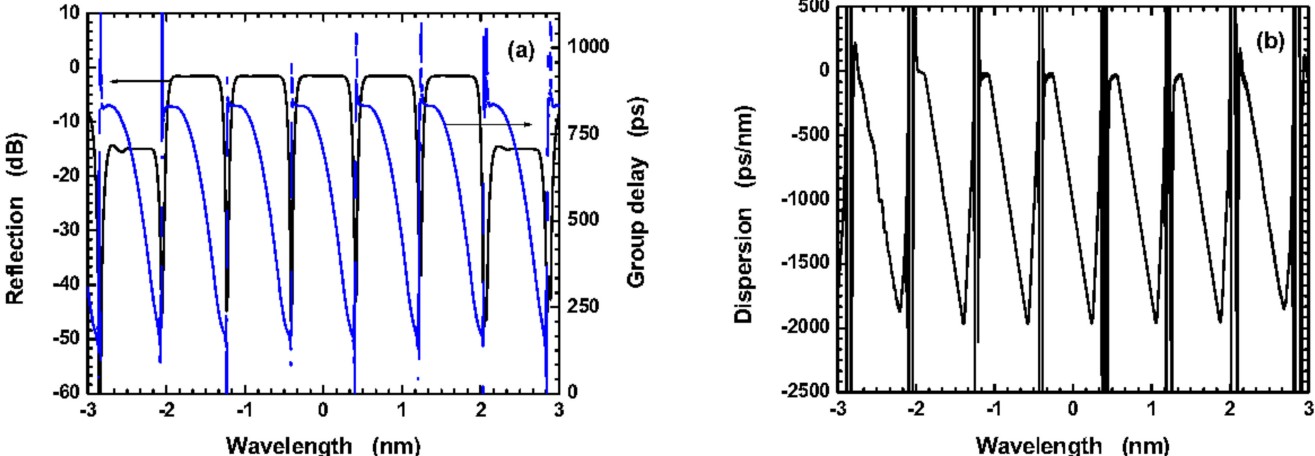

**Figure 22.** Measurement results of five-channel nonlinearly chirped FBG. (**a**) Reflection and group-delay spectrum. (**b**) Dispersion spectrum. Adapted with permission from ref. [107]. © [2022] IEEE.

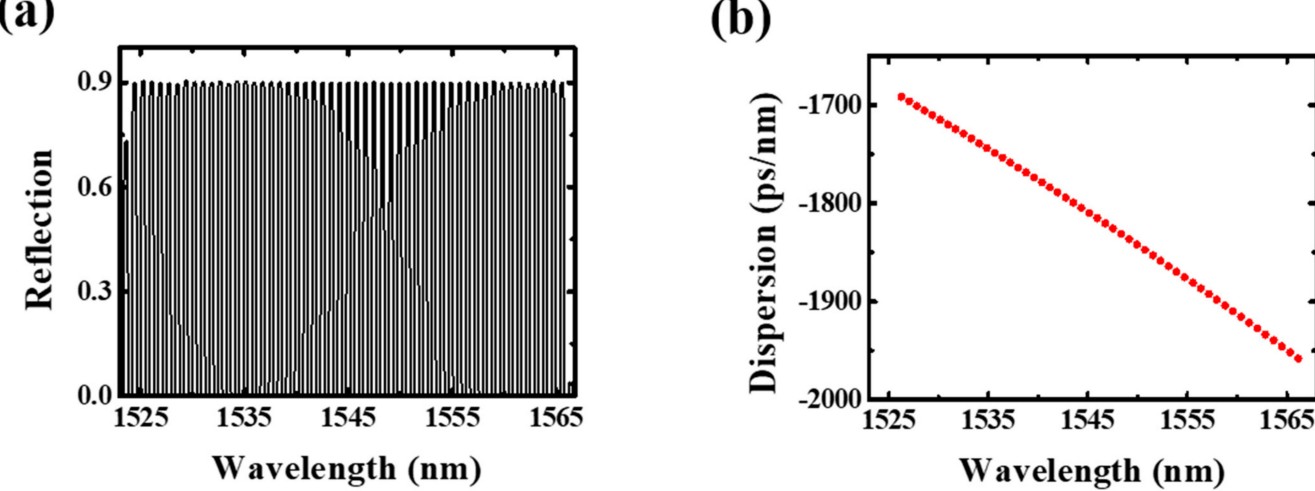

**Figure 23.** Design results for a 51-channel FBG-based dispersion and dispersion compensator. (**a**) Reflection spectrum of the 51-channel. (**b**) Dispersion spectrum of the 51-channel. Adapted with permission from ref. [115]. © The Optical Society.

Besides the above applications, the phase-only sampled FBG have also been proposed and used in the field of optical computing. Li et al., proposed a multichannel FBG-based arbitrary-order photonic differentiator [125], in which the seed grating was designed by using the discrete layer peeling algorithm (DLP) and meanwhile a multichannel was produced by using phase-only sampling method. Figure 24 shows their design results for a 45-channel first-order temporal differentiator, where the channel spacing is about 100 GHz and energy efficiency of the proposed differentiator with a bandwidth of 75 GHz is about 5.4%. By using the same design approach, the authors demonstrated the second-order photonic temporal differentiator also in [125].

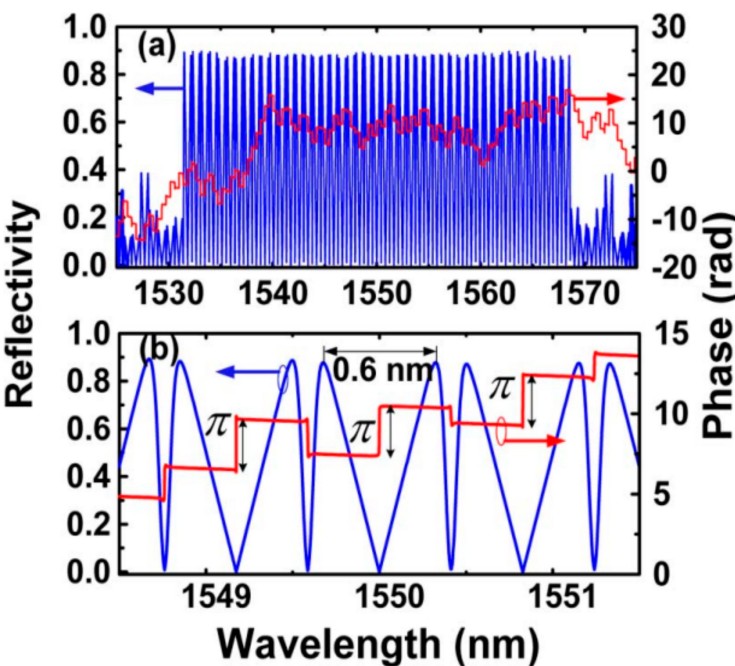

**Figure 24.** The designed results of the 45-channel first-order temporal differentiator based on a phase-only sampled FBG. (**a**) The full-view of the 45 channels, and (**b**) the zoom-in view of the central three channels. Adapted with permission from ref. [125]. © [2022] IEEE.

### 2.3. Phase-Modulated FBGs and Their Applications

The phase-modulated FBGs refer to the FBGs where there exists a continuous modulation (either linear or nonlinear ones) on the local phase of the grating itself. In general, the index modulation for the phase-modulated FBG can also be expressed by Equation (3), but there the phase-only function $s(z)$ is not a periodic one. The most typical example of such a kind of grating could be the linearly chirped FBG, where in fact the linear change in local period can be equivalently expressed as the phase of the function $s(z)$ [5]. Similar to the case of the phase-only sampled FBG, the phase distributions could be either discrete or continuous ones, and in fabrication the continuous phase distribution can be discretely encoded into each local period of grating itself.

Such a kind of FBG has two unique advantages over the amplitude-modulated FBGs: the high degree of freedom to design and obtain any customer-demanded spectrum and the fabrication feasibility due to elimination of the complex apodization in grating's amplitude. To date, the phase-modulated FBGs have found many applications and been used as, e.g., the high-repetition rate optical pulse generator [126–129], broadband dispersion compensators [123,130], arbitrary-order photonic differentiators [131], arbitrary-order photonic Hilbert [132], RZ-OOK to NRZ-OOK format converter [133], phonic millimeter wave generator [134], and linearly chirped FBG-based various sensors [135], etc.

In 2000, Longhi et al. [127] firstly proposed and demonstrated an all-optical pulse-train multiplication system, which is shown in Figure 25a, where a linearly and intensively chirped FBG with spectra (including the reflection and the group delay spectra) such as the ones shown in Figure 25b was used as a temporal multiplication component (like the time lens), the principle of which is based on the temporal Talbot effect. The experimental results are shown in Figure 25c, from which it can be seen that a 40-GHz pulse train with a pulse-width of 10 ps was successfully generated from a 2.5 GHz initial pulse train, indicating that 16-fold multiplication to the incident pulse-train can be obtained by using the proposed setup. However, as a critical disadvantage, the proposed technique generally requires ultralong linearly chirped FBGs (with length of the order of 100 cm) and, moreover, are only available to a sequence of the input optical pulses, i.e., it cannot be applied to a single pulse. To solve these two critical issues, Azana et al., proposed and demonstrated another

efficient and robust method for all-optical pulse multiplication [128], where instead of the commonly used amplitude filter, a superimposed linearly chirped FBG was especially used as a periodical phase-only filter. As a result, optical pulse burst with repetition rates as high as 170 GHz was successfully obtained from even single pulse.

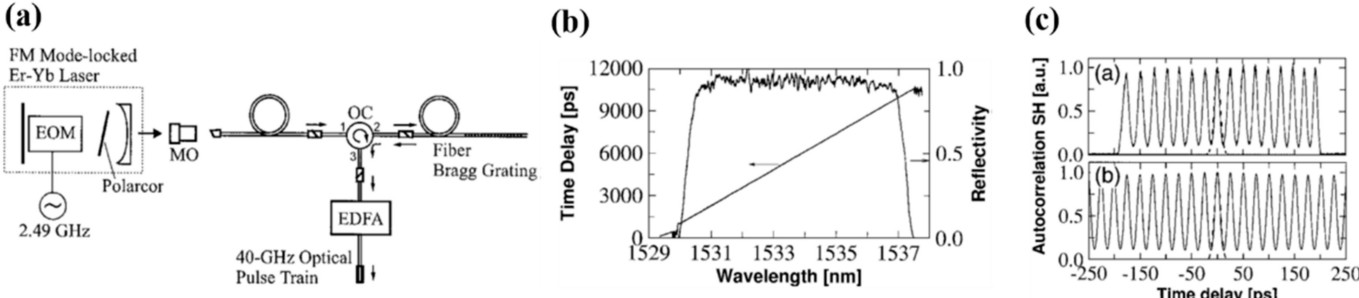

**Figure 25.** (**a**) Schematic of experimental layout for pulse–train multiplication. (**b**) Reflectivity and group delay spectra of the utilized linearly chirped FBG. (**c**) Autocorrelation trace of the multiplied pulse train (solid curves) and the incident pulse (dashed curve), in which there is (**a**) the measured result and (**b**) the simulation one. Adapted with permission from ref. [127]. © The Optical Society.

All-optical computing could be another important field for the phase-modulated FBGs [131,132]. In 2017, Liu et al., proposed and numerically demonstrated arbitrary-order temporal differentiator, which was realized by using a special phase-modulated FBG but operated in its transmission [131]. As simulation results, 0.5th-order, first-order, and second-order differentiators have been successfully demonstrated. The proposed FBG-based differentiators exhibit an excellent accuracy within a band up to 500 GHz.

Most recently, an arbitrary order photonic Hilbert converter has been proposed and demonstrated by Li et al. [132], which was realized by using an optimized phase-modulated FBG. The period and amplitude distribution of the phase-modulated FBG required for realizing the 1.5th order Hilbert transform are given in Figure 26a. The target and the simulated transmission spectrum are shown in Figure 26b. Ideally, such a kind of Hilbert converter can be expected to be obtained as long as the local period of the FBG can be precisely controlled according to those ones shown in Figure 26a.

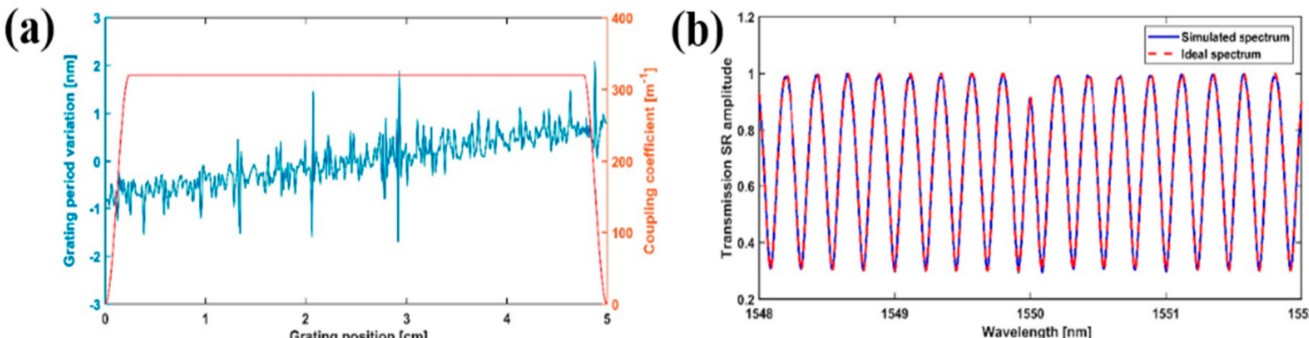

**Figure 26.** Optimization results of the 1.5th order Hilbert transformer. (**a**) The local change in grating period and coupling coefficient. (**b**) Ideal spectrum (dotted red line) and calculated spectrum. Adapted with permission from ref. [132].

In addition to the above applications, the phase-modulated FBGs have also found applications in the field of photonic microwaves. In 2008, Wang et al., proposed and demonstrated a chirped millimeter-wave pulse generator, which was realized in the optical domain with the aid of a nonlinearly chirped FBG as shown in Figure 27 [134]. The principle of the proposed approach is based on the spectral shaping (realized through a two-tap Sagnac loop filter as shown in Figure 27b,c) and nonlinear frequency–to–time mapping real-

ized by using the nonlinearly chirped FBG), i.e., the nonlinearly chirped FBG is particularly served as a strongly dispersive device enabling performing the frequency-to-time mapping. As a typical result, the photonic generation of millimeter-wave pulses with a central frequency of 35 GHz and instantaneous frequency chirp rates of 0.053 and 0.074 GHz/ps were successfully demonstrated.

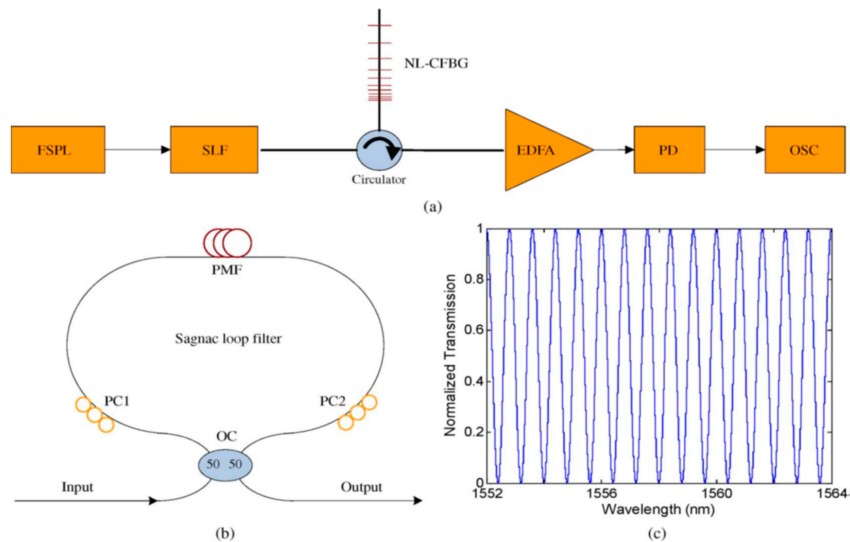

**Figure 27.** Experimental setup of the proposed chirped millimeter-wave pulse generation system. (**a**) System configuration. (**b**) Two-tap SLF. (**c**) Normalized transmission response of the SLF. Adapted with permission from ref. [134]. © [2022] IEEE.

Last but not least, to date, the phase-modulated FBGs, especially the chirped FBGs, have been profoundly studied and have found widespread applications to the fiber sensors. In this paper, we would not like to talk about this topic further. Instead, the more details and more comprehensive reviews about the versatile applications of the chirped FBGs could be found in many other review papers, such as in [135].

## 3. Phase-Inserted Long-Period Fiber Grating
### 3.1. Phase-Shifted LPGs and Their Applications

LPG refers to the fiber gratings whose periods are in the ones ranging from hundreds of micrometers to several millimeters. As a result, the resonant couplings occur only between co-propagating modes. In general, LPG can be used as an optical band-rejection filter with a bandwidth of several tens of nanometers, which is almost two orders broader than that of FBGs [136].

The phase-shifted LPG (PS-LPG) is one typical kind of LPGs and was first demonstrated by Bakhti et al., in 1997 [137], where several $\pi$ phase-shifts were purposely inserted into an LPG. As a result, a broad band-pass filter was experimentally demonstrated in the first time. Since then, the phase-shifted LPGs have been profoundly studied and have found countless applications in the fields of fiber sensors, fiber communications, and all-optical signal processing [137–184], etc.

3.1.1. Spectral Characteristics and Fabrication Techniques for the Phase-Shifted LPG

In 1998, Ke et al., systematically analyzed the spectral characteristics of the phase-shifted LPG (PS-LPG) [138]. They numerically validated the previous results reported in [137] and revealed, for the first time, that some unusual spectral-responses, such as the double notches or a flat-top notch, can easily be obtained by using the PS-LPG where the number and quantities of the inserted phase-shifts are suitably selected [138]. In 1999, when Liu et al., performed analyses on PS-LPGs and the cascaded LPGs together, they further revealed that the transmission spectrum of the PS-LPG could be flexibly tailored by adjusting

the numbers, the inserted positions, and quantities of the inserted phase-shifts [139]. Since then, PS-LPGs have frequently been used to design and realize some complex filters such as the flat-top bandpass filter [140], the flat-top band-rejection filter [141], the gain-flattening filter [142,145,166,167], the twin broadband rejection filter [143,144], etc., which in general cannot be realized by using the conventional LPGs.

On the other hand, referring to fabrication techniques of the PS-LPGs, various methods have been developed to date. Similar to those PS-FBGs which have permanent phase-shifts, the PS-LPGs can mainly be classified by two categories, i.e., the post-processed ones and the directly generated ones. For the former category, the phase-shifts are generated and inserted into LPG after the LPG is fabricated, which includes, for example, the UV post-processing ones [145,146]; the post-etching ones [147,148]; the post-tapering ones based on either the electric arc discharge [149], the filament heating [150], or the $CO_2$ laser exposure ones [151]; and the mechanical pressure one [152], etc. These post-processing methods enable inserting the phase-shifts into the LPG with great flexibility. However, all these methods need either the cost-ineffective UV writing system or very complicated and time-consuming etching and coating procedures. Moreover, by using the above methods, it is extremely difficult to precisely control the magnitudes and insertion position of the phase-shifts.

For the latter category, the required phase-shifts are created and formed in LPG by directly inserting additional non-grating (bare fiber) regions at the specified positions of the grating during its fabrication, which mainly include the point-by-point directing writing methods based on electric arc discharge [153,154] and the $CO_2$ laser [155–159], etc. Such methods enable the robust yet precise fabrication of the PS-LPGs and thus have attracted increasingly a great interest in recent years. Of these, the $CO_2$ laser-based point-by-point grating writing techniques have attracted especial attention [155–159] due to their simplicity and cost efficiency in the fabrication system where there are no limitations for fiber selection, i.e., even the conventional SMF fiber can be used to fabricate LPGs.

Back in 2005, Zhu et al., first reported and demonstrated the point-by-point $CO_2$ laser technique for the fabrication of the PS-LPG [155], where the LPG was generated by periodically deforming the fiber surface through the focused $CO_2$ laser beam, and meanwhile the phase-shift was introduced by inserting an additional blank region at center of the grating, which are shown in the insets of Figure 28. Two symmetrical notches with rejection depths of ~11 dB were experimentally obtained, as shown in Figure 28, which are exactly the same as those previously expected and experimentally demonstrated in [137,138].

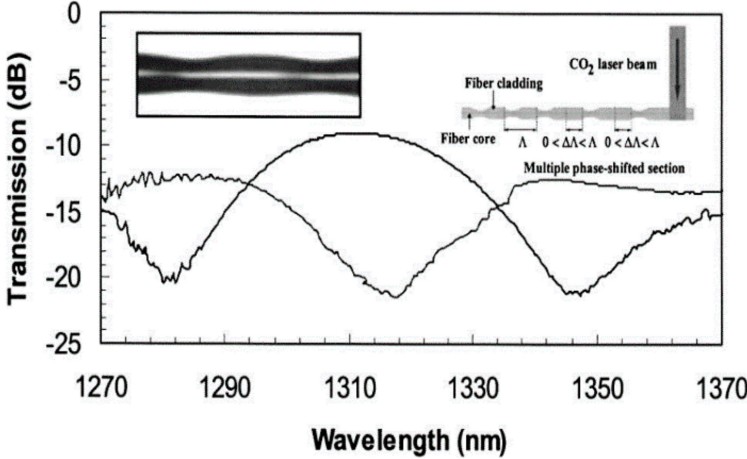

**Figure 28.** Transmission spectra of the LPG with (solid curve) and without (gray curve) the p-phase-shift at the grating center. Insets, (left) microphotography of the PS-LPG surface and (right) schematic diagram for formation of the PS-LPG. Adapted with permission from ref. [155]. © The Optical Society.

Since then, such point-by-point $CO_2$ laser techniques have widely been used to fabricate different kinds of the PS-LPGs in either SMFs [156–160] or photonic crystal fibers (PCFs) [161]. Of these, the multiple phase-shifted helical long-period fiber grating (HLPG) was firstly proposed and experimentally demonstrated by Gao et al. [160], where two particular phase-shifts, i.e., $\pi/2$ and $3\pi/2$, were inserted into an HLPG by just changing two local periods.

HLPG refers to a piece of fiber where there exists spiral-type index modulation distribution along the fiber axis. As a new kind of LPG, HLPGs have recently attracted a great research interest and have been found numerous applications, such as orbital angular momentum (OAM) beam convertors, torsion sensors, polarization-insensitive band-rejection filters, and circular polarization convertors, etc. [162,163]. In 2014, the author Li's group developed a novel fiber-twisting technique for the fabrication of the HLPG [164] and the phase-shifted HLPG (PS-HLPG) [165], the schematic diagram of the fabrication setup is shown in Figure 29a, where a sapphire tube is particularly utilized in place of the focal lens. The period of the HLPG is precisely controlled by adjusting the speeds of both the fiber-moving stage and the rotator. Since the sapphire tube rather than the fiber is directly heated by the $CO_2$ laser, the passed fiber within the tube region is homogeneously heated, and thus no deforms can be found in the fiber surface. The principal schemes for fabricating the HLPG and the PS-HLPG are shown in Figure 29b, where the inset (I) shows a PS-HLPG with a phase-shift of $\theta$, whereas the inset (II) shows the HLPG without any phase-shifts insertion. Here it must be noted that the phase-shift $\theta$ is an accumulated phase, which can be regarded as the phase inserted right at end of the part $L_2$. Therefore, an arbitrary phase-shift can be obtained just by changing the length of $(L_2 - L_1)$. Unlike those post-processing methods in which the phase-shift is formed after the fabrication of LPG is completed, here both the HLPG and the inserted phase-shift are formed simultaneously.

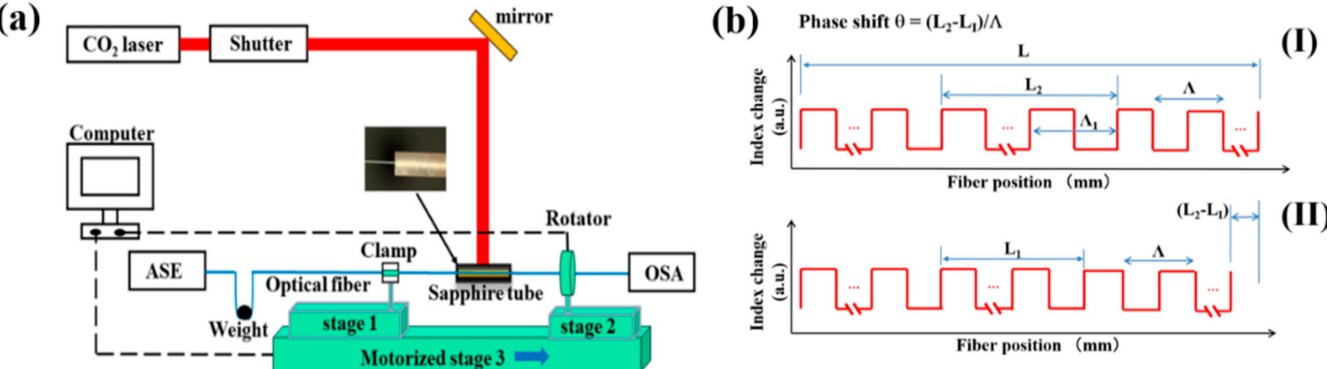

**Figure 29.** (**a**) Sapphire tube and $CO_2$ laser-based processing platform for HLPG fabrication. (**b**) Principal scheme for formation of a phase-shifted HLPG. Adapted with permission from ref. [165]. © The Optical Society.

### 3.1.2. Applications of the PS-LPGs to All-Fiber Optical Filters and All-Optical Computing Devices

With progress on theoretical works of the PS-LPGs [140–144], many efforts to explore the practical applications of the PS-LPGs to the fields of fiber optics and all-optical computing have also been accomplished. To date, the PS-LPGs have been used as dual-band rejection filters [143,161], gain-flattening filters [145,166,167], optical differentiators [168–170], spectral filters in fiber lasers [171,172], and multiple orbital angular momentum (OAM) mode converters [173], etc.

Of these, gain-flattening filter used for EDFA could be one of the most important applications of the PS-LPGs to the fiber optics. In 2002, Harumoto et al., proposed and demonstrated experimentally a gain-flattening filter for EDFA, which was realized based on a utilization of a PS-LPG in combination with a conventional LPG [167]. In the results, a gain-flattening filter with a bandwidth of 37 nm (covering the full C-band) and gain non-uniformity of less than 0.45 dB was successfully obtained.

The optical differentiator is another typical application example of the PS-LPGs. Attributed to the intrinsic broadband characteristics of the LPGs, the LPG-based optical differentiator is available to a temporal waveform with a bandwidth up to several THz, which is almost two orders broader than that of the FBG-based ones. In 2005, by using a π PS-LPG combined with a uniform LPG, Azana et al., demonstrated numerically a first-order optical differentiator [168]. Later, Kulishov et al. [169] and Ashrafi et al. [170] further proposed and numerically validated that the Nth order differentiator could also be obtained by using an N−1 π PS-LPG.

Besides the above application examples, the PS-LPGs, acted as intra-cavity filters, have also been used in fiber lasers [171,172]. Jiao et al., proposed and demonstrated a method to narrow the spectral linewidth of a fiber laser [172], where a π PS-LPG was inserted into the laser cavity and especially used to suppress the spectral broaden effects resulting from the nonlinear optical effects (including the self-phase modulation and the four-waves mixing effects).

Most recently, the PS-LPGs have also been proposed and used for 2 × 2 switching in a multi-wavelength multiple OAM mode system [173], where two cascaded PS-LPGs were utilized as an optical switch, which enables mechanically exchanging the zero-order OAM mode and the second-order OAM mode with each other at two different wavelengths, as shown in Figure 30a, whereas the intensity and phase distributions of the output OAM mode are shown in Figure 30b. It can be seen that 2 × 2 wavelength switchings with power efficiencies of ∼98.4% and ∼98.7%, respectively, for the two OAM modes are successfully demonstrated at wavelengths of 1537 and 1558 nm, respectively.

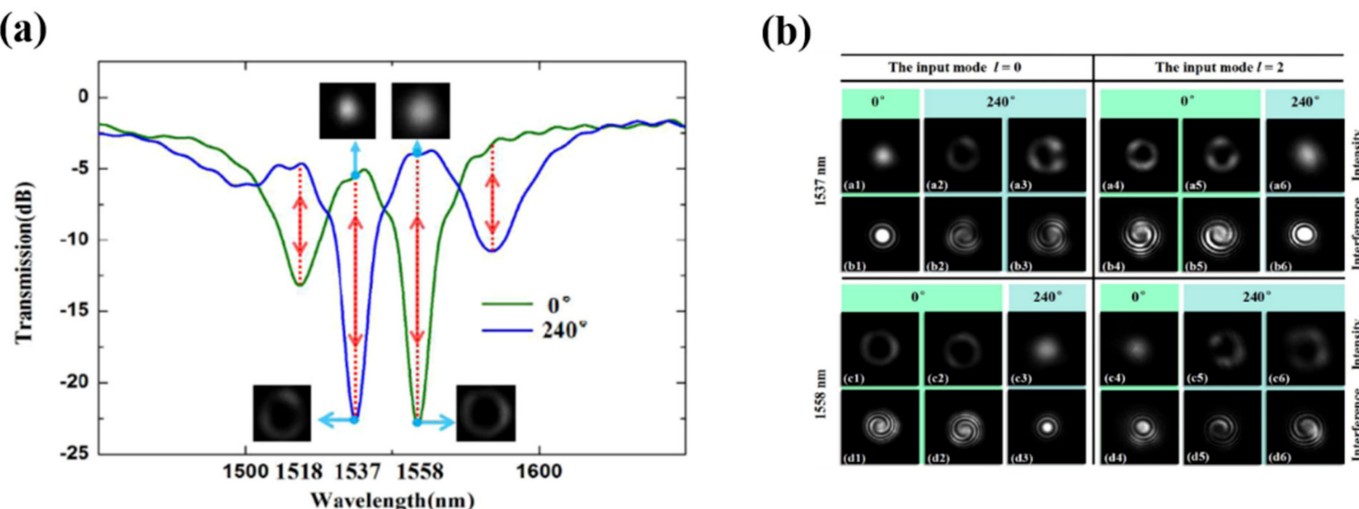

**Figure 30.** (**a**) Changes of the resonant mode at different wavelength under two different torsions. (**b**) Intensity and phase distributions of output modes. Adapted with permission from ref. [173]. © The Optical Society.

### 3.1.3. Applications of the PS-LPGs to Fiber Sensors

Compared with their applications in other fields, applications of the PS-LPGs in fiber sensors have attracted more attention and acquired more developments. To date, various kinds of the PS-LPG-based sensors have been proposed and demonstrated, which include, e.g., the bending sensors [146], transverse load (force) sensors [149], strain sensors [150,159],

refractometric sensors [151,154], temperature sensors [174], and torsion sensors [160,174], etc.

In 2013, the author Li's group proposed and demonstrated a sensing structure enabling the simultaneous measurements of temperature and the ambient refractive index [151], which was realized by using an equivalent PS-LPG, i.e., the PS-LPG was formed by tapering an LPG on its central region. Figure 31a shows the schematic diagram of the PS-LPG. Figure 31b shows the measurement results for the dependence of the wavelength on refractive index of the ambient solvent within the tapered region.

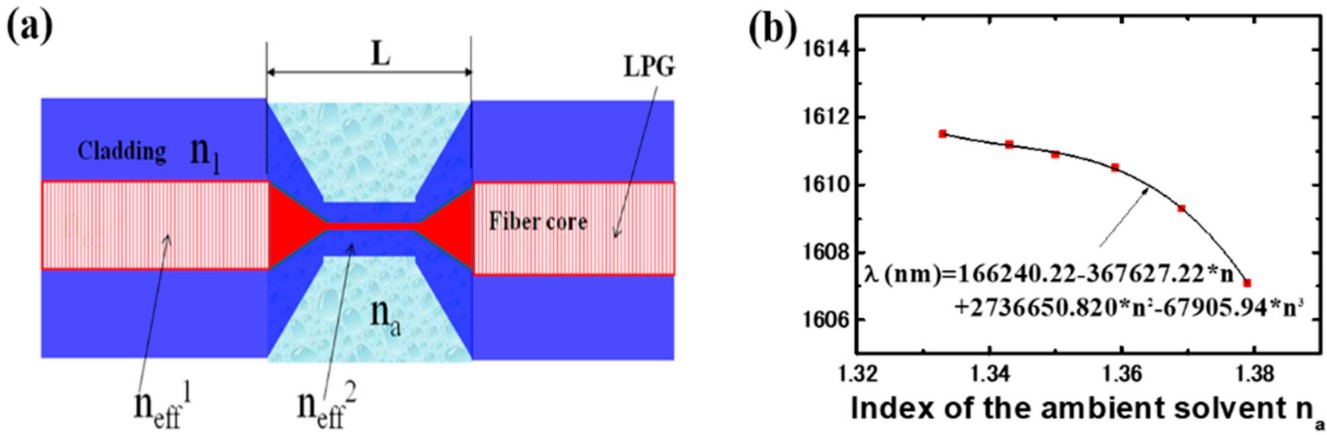

**Figure 31.** (**a**) Schematic diagram of the tapered fiber-based phase-shifted LPG. (**b**) Dependent of the peak wavelength on the refractive index of the ambient solvent. Adapted with permission from ref. [151]. © The Optical Society.

The LPG-based torsion sensor, especially the HLPG-based torsion sensor, is another important topic of research that has recently attracted a great interest [164]. It has been validated that, compared with the LPG-based torsion sensors, the HLPG-based torsion sensors have 5–10 times higher torsion responsivity. More importantly, the torsion direction could be discerned by using only one HLPG itself, which makes the HLPGs especially suitable to the fiber-based torsion sensors [163]. In 2013, Gao et al., proposed and demonstrated a temperature-insensitive torsion sensor [160], which was realized by using a PS-HLPG but with dual phase-shifts ($\pi/2$ and a $3\pi/2$) insertion, as shown in Figure 32, where Figure 32a,b show the schematic diagram and transmission spectra of the proposed PS-HLPG, respectively. From Figure 32a, it can be seen that a $\pi/2$ and a $3\pi/2$ phase-shift were introduced in the HLPG. Attributed to the insertion of these two phase-shifts, two new peaks in transmission spectrum appear on both sides of the original peak, as shown in Figure 32b. With changes in temperature and the applied torsions, wavelengths of all three peaks will be shifted to either the short wavelength or the long wavelength direction, but the wavelength difference among these three peaks remain constant whenever the temperature is changed. As a result, a temperature-insensitive torsion with a torsion sensitivity up to 1.959 nm/(rad/m) was successfully demonstrated.

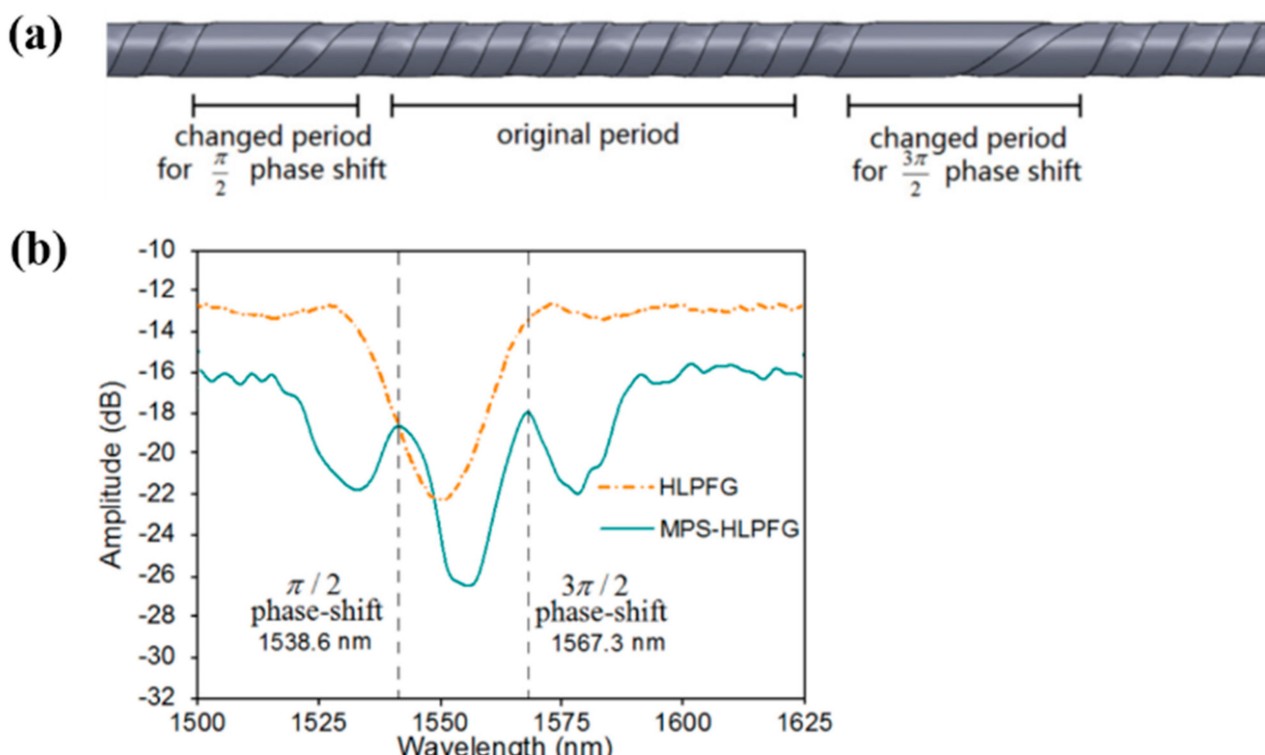

**Figure 32.** (**a**) Schematic diagram and (**b**) transmission spectra of the proposed PS-HLPG. Adapted with permission from ref. [160]. © The Optical Society.

*3.2. Phase-Only Sampled LPGs/HLPGs and Their Applications*

Before the year of 2018, most of the research works on LPGs/HLPGs were mainly limited to the single-channel one, i.e., there exists only one spectral notch for each of the coupled cladding mode; multi-channel LPGs/HLPGs were rarely studied and demonstrated, which, however, are essential to wavelength-division multiplexing (WDM) devices in both the fiber sensing and fiber communication systems. Most recently, the author's group has proposed and demonstrated a phase-only sampled nine-channel HLPG [175], which was the first time for the practical demonstration of multichannel HLPGs in the world. To facilitate the fabrication such kinds of LPGs/HLPGs by using the $CO_2$ laser technique, the author's group has further proposed and demonstrated a three-channel LPG and a five-channel HLPG, where instead of the conventional sampling method (i.e., the sampling function is generally added to modulate the AC part of the seed grating), either a helical [176] or a rectangular [177] sampling function was utilized to modulate the direct current (DC) part of the seed LPG/HLPG. In 2021, the author's group further proposed and demonstrated an equivalent phase-only sampling method to realize multichannel HLPG [178], where instead of inscribing the discrete phases into each local period of the seed HLPG, the phase-only sampling function is realized by superposing the sampling HLPGs with the seed HLPG in a same region of the fiber. The proposed approach would greatly facilitate the fabrication processes of the multichannel HLPG by using the direct $CO_2$ laser writing technique; meanwhile, the proposed method enables providing many more flexibilities in the design and fabrication of high-channel-count HLPG than the previous methods.

3.2.1. Design Principle and Fabrication Results for the Phase-Only Sampled HLPG

In general, the refractive index modulation in a phase-only sampled HLPG $\Delta n_M(r,z,\varphi)$ can be expressed as [175,178]

$$\Delta n_M(r,z,\varphi) = \mathrm{Re}\{\Delta n_s(r,z,\varphi) \cdot S(z)\} = \mathrm{Re}\left\{ \frac{\Delta n_1(r,z,\varphi)}{2} \cdot \exp\left(i\sigma\frac{2\pi}{\Lambda_0}z\right) \cdot s(z) \right\} \quad (9)$$

where $r$, $\phi$, and $z$ are the radial position, the azimuthal angle, and the axial position along the HLPG, respectively. $\Delta n_s(r, z, \varphi)$ represents the index modulation of the seed HLPG (i.e., the uniform HLPG). The sign Re denotes the real part of the complex number in bracket. $\Delta n_1(r, z, \varphi)$ and $\Lambda_0$ represent the maximum index modulation and central pitch of the seed HLPG, respectively. The variable $\phi$ is a function of the position $z$, which satisfies the equation $\phi(z) = \phi(z + N \cdot \Lambda_0)$ (N is an arbitrary integer). $\sigma$ represents the handedness and helix of the utilized HLPG, $\sigma = +1$ and $\sigma = -1$ represent the left-handed and right-handed, respectively, and the helix is one. $s(z)$ denotes a continuous phase-only sampling function, which is exactly identical to the one what we defined and used in Equations (4) and (7). The procedures to optically design the phase-only sampling function are exactly the same as what we have performed for phase-only sampled FBG. Figure 33 shows the optimization result for phase distributions of the three- and nine-channel phase-only sampling function.

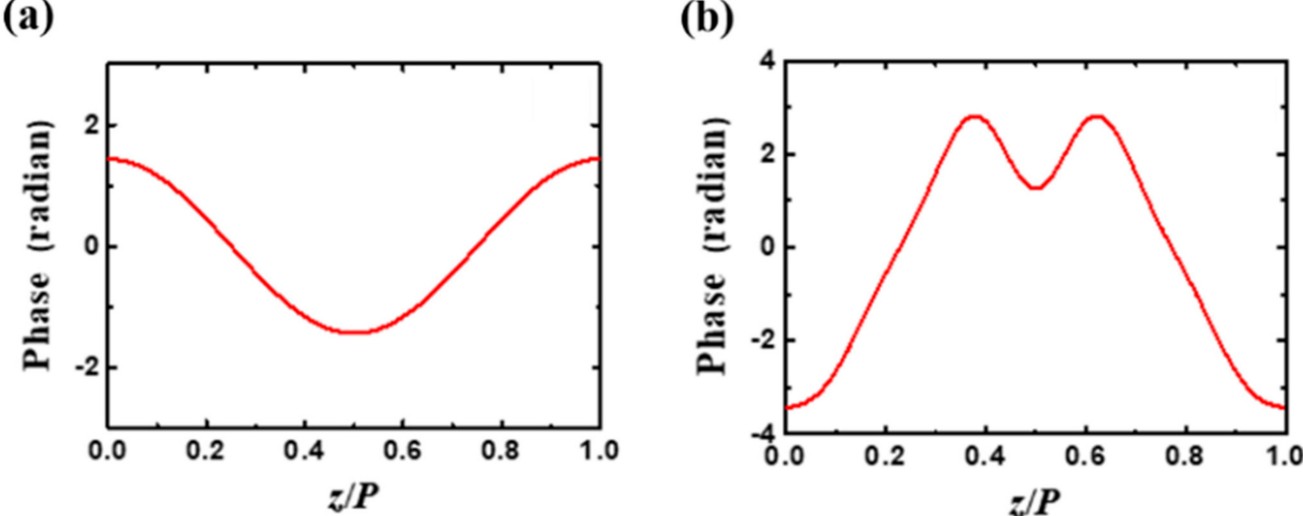

**Figure 33.** Optimized results for phase distribution of the phase-only sampling function: (**a**) 3-channel and (**b**) 9-channel. Adapted with permission from ref. [175]. © The Optical Society.

The simulation results for spectra of the three-channel HLPG and nine-channel HLPG are shown in Figure 34a,b, respectively. In addition, these two designed HLPGs have been fabricated by using the sapphire tube- and $CO_2$ laser-based processing platforms, as shown in Figure 29a. Figure 34c,d show the measurement results for the transmission spectra of the two fabricated HLPGs, respectively. All the results indicate that, as expected, three-channel and nine-channel HLPGs have been successfully obtained [175].

Instead of the aforementioned phase-sampling method where the continuous phases of the phase-only sampling function are discretely encoded into each local period of the seed HLPG, the author's group has further proposed and demonstrated an equivalent phase-sampling approach [176–178]. The proposed method relies on the fact that in most of the cases, the sampling period $P$ must be much larger than that of seed grating; therefore, the phase distribution of the sampling function can be equivalently obtained by using a new sampling function, namely the DC sampling one, to modulate the DC part of the HLPG's index modulation. Equation (9) then can be equivalently rewritten as [178] the following:

$$\Delta n_M \approx \Delta n_s + \Delta n_{DC} \tag{10}$$

where $\Delta n_{DC}$ represents the newly resulted DC part of the index modulation endured by the seed HLPG, which can easily be obtained by using the Equation (7) and mathematically expressed as

$$\Delta n_{DC} = -\frac{\lambda}{2\pi} \frac{d\theta(z)}{dz} = \frac{\lambda}{P} \sum_{m=1}^{J} m a_m \sin\left(\frac{2\pi m z}{P}\right) = \sum_{m=1}^{J} \Delta n_2(m) \sin\left(\frac{2\pi m z}{P}\right) \tag{11}$$

where λ is the central wavelength. The DC-part index modulation can be equivalently regarded as the linear summation of the *J* gratings, and each of these gratings has its period of $P/m$ and its amplitude of $\Delta n_2(m) = m\alpha_m\lambda/P$ in terms of the different numbers of *m*. By using the DC sampling method, the author Li's group demonstrated three-channel HLPG both theoretically and experimentally [176,177]. Figure 35 shows the schematic structure of the equivalent phase-only sampled HLPG, where the seed HLPG and a binary sampling grating (called the DC sampling function) are superimposed each other within the same fiber region [177].

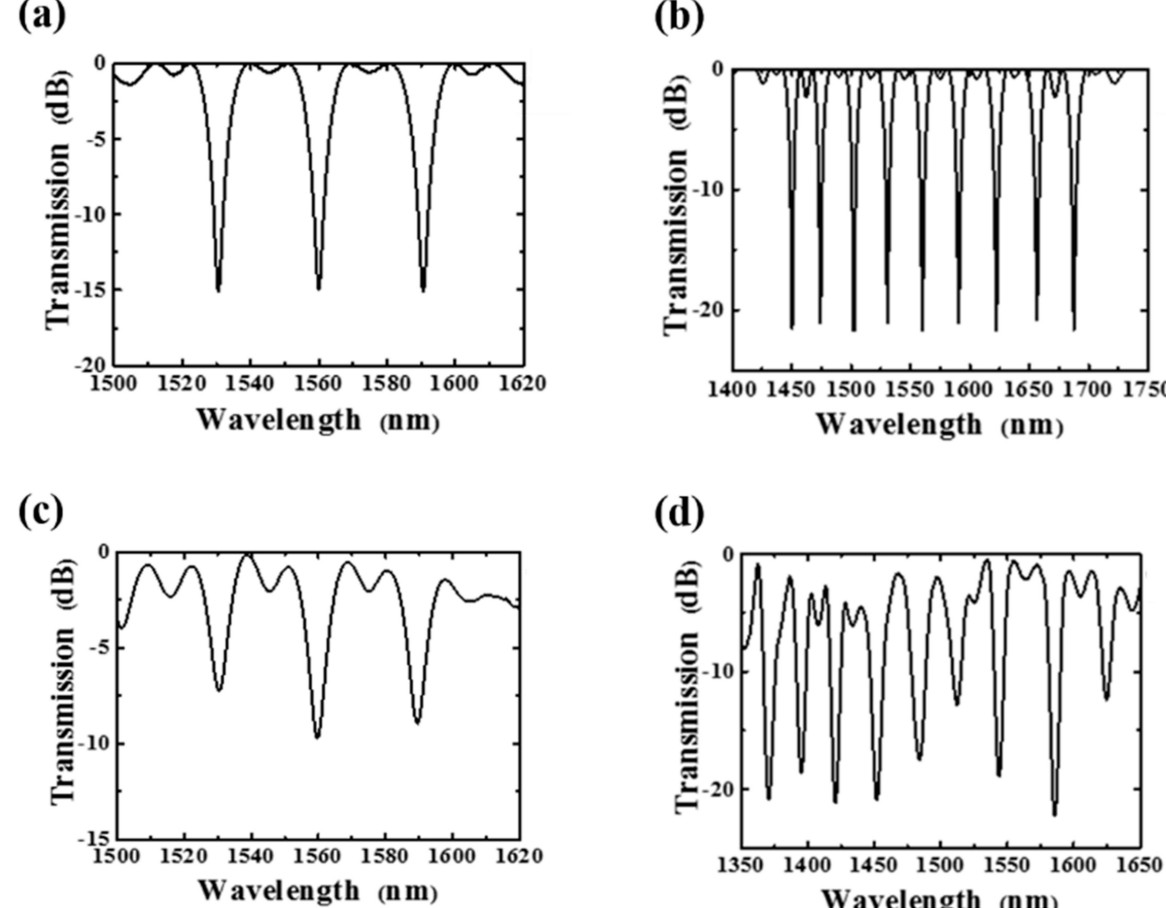

**Figure 34.** Design and experimental results for 3- and 9-channel phase-only sampled HLPG. Design results for (**a**) 3-channel and (**b**) 9-channel HLPGs. Experimental results for (**c**) 3-channel and (**d**) 9-channel HLPGs. Adapted with permission from ref. [175]. © The Optical Society.

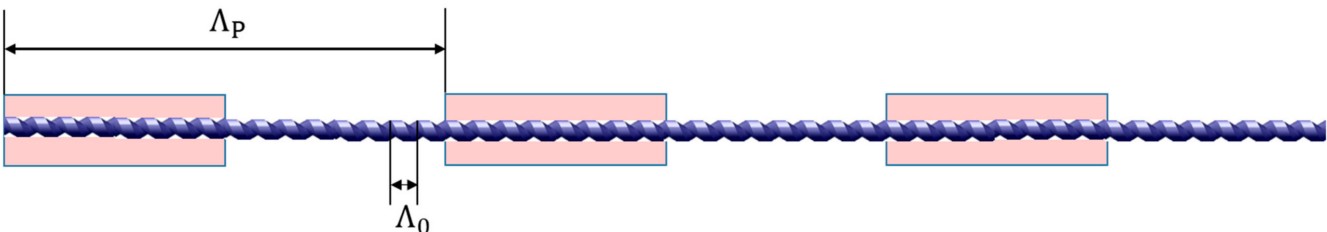

**Figure 35.** Schematic structure for the proposed DC-sampled HLPG. Adapted with permission from ref. [177]. © [2022] IEEE.

To further increase channel count of the DC-sampled HLPG, most recently, the author Li's group has proposed and demonstrated a novel method (called the superimposed HLPGs) to produce the multi-channel HLPG [178]. Figure 36 shows the schematic diagram

of the conceptual principle, where the HLPG represents the seed grating, whereas the HLPG #1-HLPG#*J* represents the HLPGs with periods of $P$, $P/2$, ..., and $P/J$, respectively. In principle, the equivalent phase sampling function with harmonic terms as shown in Equation (11) can be realized by superimposing each of the sampling HLPGs in turn on the seed HLPG. For example, a three-channel HLPG can be obtained by linearly superimposing the seed HLPG with the HLPG#1, whereas a five-channel HLPG can be obtained by linearly superimposing the seed HLPG with two DC sampling HLPGs, i.e., the HLPG#1 and the HLPG#2 but with different weight coefficients [178].

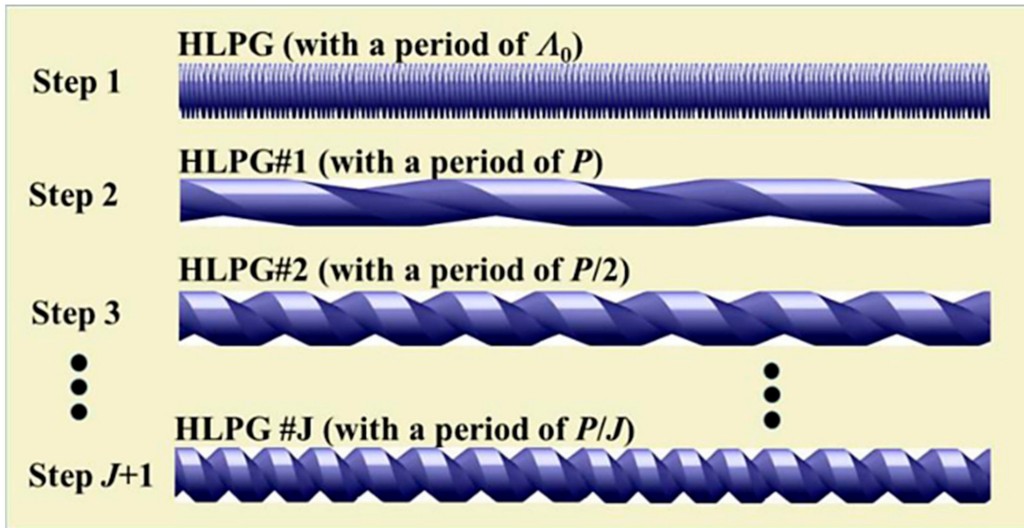

**Figure 36.** Schematic diagram of the superimposed HLPG. Adapted with permission from ref. [178]. © [2022] IEEE.

### 3.2.2. Applications of the Phase-Only Sampled HLPG/LPG

Owing to the unique performance of the high uniformity in both the inter- and the intra-channels, the phase-only sampled HLPGs/LPGs can be used as a multi-channel band-pass/rejection filter, which may find applications in the fields of fiber communication, all-optical computing, and fiber sensors.

On the other hand, it has recently been demonstrated both theoretically and experimentally that the higher azimuthal modes existing in the HLPGs are of the inherent OAM modes [179], which implicitly means that the phase-only sampled HLPGs can be used as multichannel OAM mode converters. In 2019, by using three-channel DC-sampled HLPG, the author's group proposed and experimentally demonstrated a three-channel OAM mode converter, which was the first report of a real demonstration of the HLPG-based multichannel OAM mode converter in an experiment [177]. Based on the utilization of the superimposed HLPGs, five-channel OAM mode generators have recently been obtained by us too [178]. Some of the typical results about the spectrum and the corresponding intensity and phase distributions of the OAM modes are shown in Figure 37, from which it can be obviously seen that three first-order OAM modes with a conversion efficiency larger than 11.8 dB (~93%) have been generated at three different wavelengths (channels), respectively.

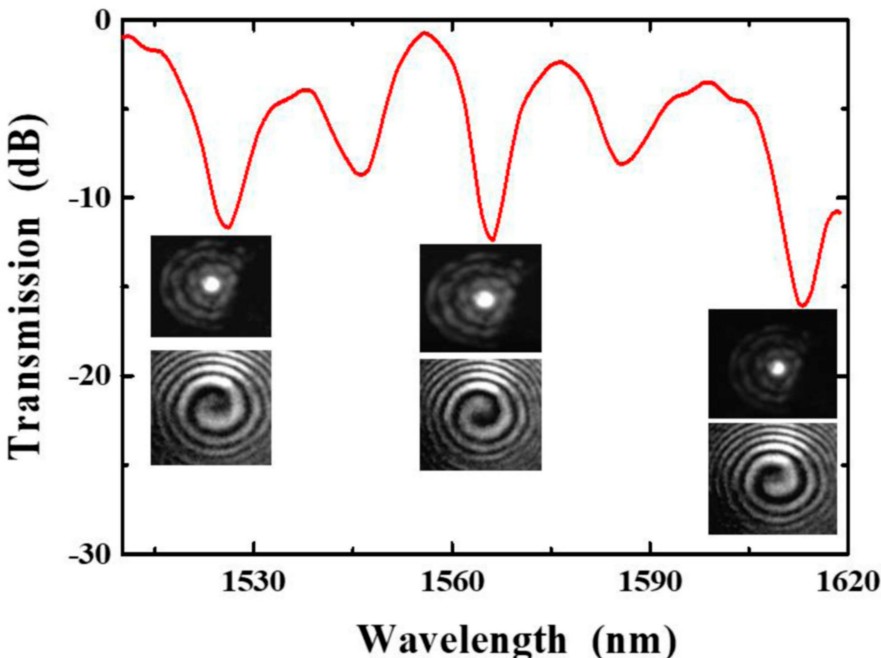

**Figure 37.** The spectrum, the corresponding intensity, and phase distributions of the OAM modes generated at three channels of the HLPG simultaneously. Adapted with permission from ref. [178]. © [2022] IEEE.

In addition to the above application, the phase-only sampled HLPG can also be used as an efficient sensing head in a multiple-parameter fiber-sensing system. Most recently, the author's group has proposed and demonstrated a new sensor for the simultaneous measurement of the directional torsion and the ambient temperature, which was realized by using a DC-sampled HLPG [180]. Figure 38a,b shows the transmission spectra of the three-channel HLPG-based torsion sensor under different torsions and temperatures, respectively. Due to the large different of mode dispersion at the central wavelengths of the three channels, i.e., wavelengths of the three dips as shown in Figure 38a,b, these three dips all have linear responses to the changes in torsions and temperatures, but the response slopes are largely different, which thus enables one to eliminate the crosstalk effect of the torsions with the temperatures by simply using the 2 × 2 matrix method [180].

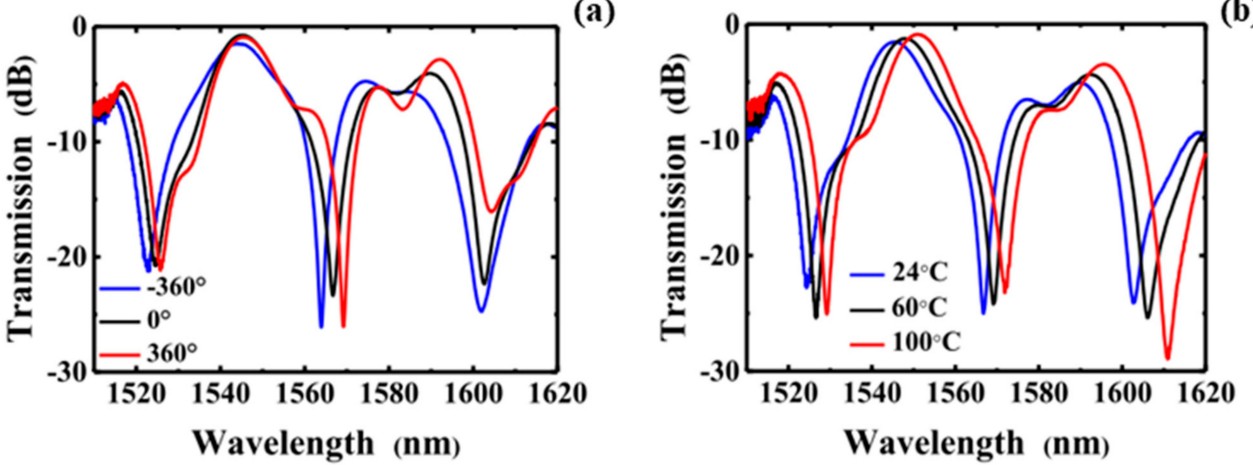

**Figure 38.** Transmission spectra of the 3-channel HLPG-based fiber sensor under different (**a**) torsion angles and (**b**) temperatures. Adapted with permission from ref. [180].

### 3.3. Phase-Modulated HLPGs and Their Applications

The phase-modulated LPG, especially the phase-modulated HLPG, can be exploited for the design and generation of the edge filter [181], broadband flat-top filter [182], and broad-band OAM mode converter [183,184], etc.

By means of a phase-modulated HLPG, the author's group has experimentally demonstrated a polarization-independent flat-top filter [182]. The proposed HLPG does not need a complex apodization in its amplitude, which considerably facilitates the fabrication processes and makes the designed HLPG particularly suitable for fabrication by using the $CO_2$ laser-writing platform. Figure 39 shows the transmission and PDL spectra of fabricated phase-modulated HLPG. From this, it can be seen that a polarization-independent band-rejection filter with a bandwidth of ~10 nm at −20 dB and a rejection depth of ~28 dB has been successfully achieved.

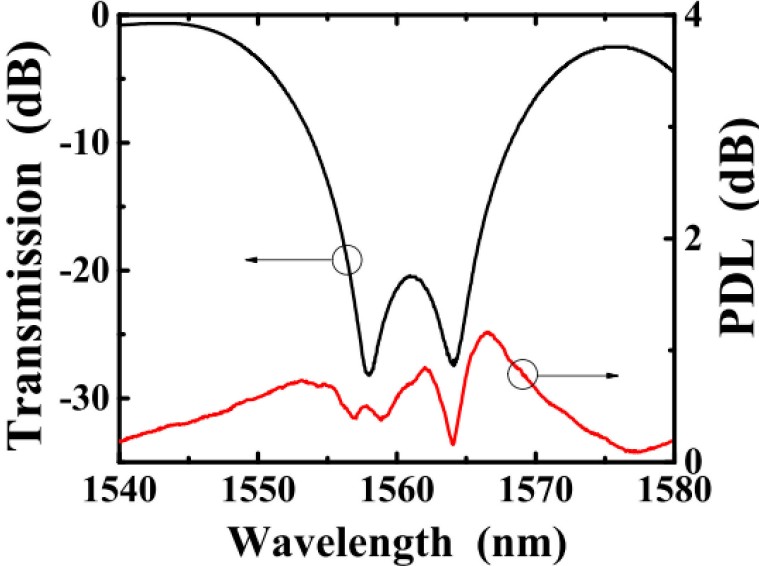

**Figure 39.** Transmission and PDL spectra of the fabricated phase-modulated HLPG. Adapted with permission from ref. [182]. © [2022] IEEE.

Based on utilization of the phase-modulated HLPG, most recently, the author's group has further proposed and numerically demonstrated a broadband flat-top second-order OAM converter, the schematic and the simulation results of which are shown in Figure 40, where the utilized HLPG is a phase-modulated one that is especially inscribed in a thinned, four-mode fiber and operated at wavelengths near the dispersion turning point (DTP). In contrast to most of the other HLPG-based OAM mode generators where the high-order OAM mode and flat-top broadband have rarely been achieved simultaneously, a second-order OAM mode converter with a flat-top bandwidth of 113 nm at −20 dB and a depth fluctuation of less than 3 dB at −26 dB has been successfully demonstrated. The obtained flat-top bandwidth covers the entire C + L bands.

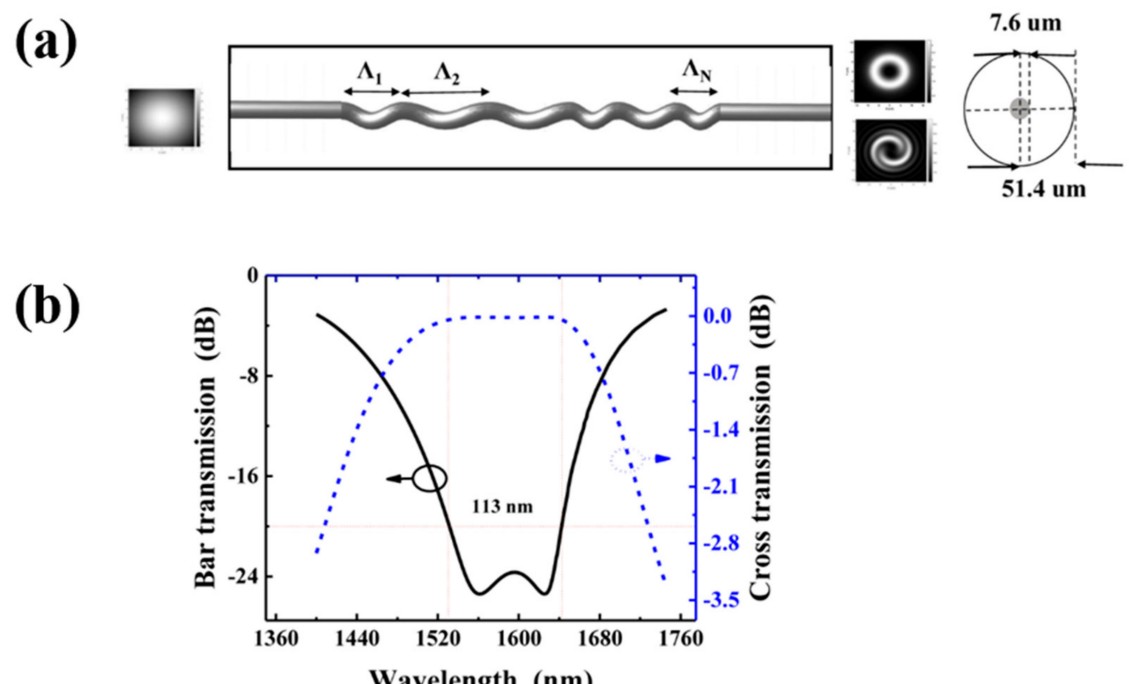

**Figure 40.** (**a**) The schematic diagram for the proposed phase-modulated HLPG inscribed in a thinned four-mode fiber. (**b**) Transmission spectrum of the proposed phase-modulated HLPG. Adapted with permission from ref. [184]. © The Optical Society.

## 4. Conclusions and Prospects

In this paper, the past and recent advances on the phase-inserted fiber gratings and their applications have been reviewed. It is shown that the PI-FGs can be divided into three categories, i.e., PS-FGs, phase-sampled FGs, and phase-modulated FGs. Principles and fabrication techniques of these three kinds of fiber gratings and their applications in the fields of optical communication, optical computing and optical signal processing, microwave photonics, and optical fiber sensors have been introduced and summarized in details.

In the past decades, the FBG-based PI-FGs, especially the PS-FBGs, have been comprehensively studied and widely used in the fields covering nearly all the optical interdisciplinary fields. Meanwhile, fabrications of the PS-FGs have also been widely studied and become established techniques to date.

In accordance with the great developments and advances obtained on FBG-based PI-FGs, the research on LPG-based PI-FGs and their applications have also gained large progresses but obviously lack comparability in both the width and the depth compared with those on PF-FGs. On the other hand, attributed to the intrinsic helicity characteristics that are especially suitable to control the loss, polarization, and OAM states of the light in optical fiber, HLPG and HLPG-based devices have recently attracted great research interest. To date, HLPGs have been comprehensively studied and have found many applications, such as in torsion sensors, OAM mode converters, broadband flat-top filters, circular polarizers, etc., and in the fields of fiber sensors, fiber communications, quantum optics, ultrahigh precision measurement, etc. The research on phase-inserted HLPGs (PI-HLPGs) is still in its infancy. Applications of the PI-HLPGs in fields such as photonics information processing, chiral photonics, quantum optics, etc., have rarely been explored and demonstrated. It is believed that PI-HLPGs including the phase-shift, phase-sampled, and phase-modulated HLPGs will be further explored and find many practical applications in the near future.

Another research trend for PI-FGs could be the PT symmetry grating. Recently, the parity-time (PT) symmetry FBG has attracted much attention, especially in the fields of

optics and photonics [185]. However, the research and applications of the phase-inserted PT symmetry gratings including the PT-HLPG/HFBG and PT-LPG have rarely been reported so far; therefore, it is believed that such kinds devices would attract more attention and find potential applications in the near future.

**Author Contributions:** Writing and Editing, C.Z., L.W. and H.L. All authors have read and agreed to the published version of the manuscript.

**Funding:** Funding has been provided by the KDDI Foundation for Research Grant Program; Yazaki Memorial Foundation for Science & Technology; JSPS KAKENHI Grant Number JP 22H01546; and National Natural Science Foundation of China (62003081).

**Institutional Review Board Statement:** Not applicable.

**Informed Consent Statement:** Not applicable.

**Data Availability Statement:** Not applicable.

**Conflicts of Interest:** The authors declare no conflict of interest.

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
