# Peer review of "Phase-Inserted Fiber Gratings and Their Applications to Optical Filtering, Optical Signal Processing, and Optical Sensing: Review"

_photonics, doi:10.3390/photonics9040271_

Round 1

Reviewer 1 Report

In this manuscript, the authors reviewed the past and recent advances on the phase-inserted fiber gratings (PI-FGs) and their applications. The PI-FGs developed to date were divided into three categories: phase-shifted gratings (PS-FGs), phase-only sampled gratings, and phase-modulated gratings, of which the utilized gratings could be either the Bragg ones (FBGs) or the long-period ones (LPGs). And the principles and fabrication techniques of these three kinds of fiber gratings, and their applications in the fields of optical communication, optical computing and optical signal processing, mi-crowave photonics, and optical fiber sensors have been introduced and summarized in details. The content of the manuscript has great timeliness. Moreover, the authors discuss the research progress of PI-FGs with excellent breadth and accuracy. The full text of the manuscript is fluent and logical, and the description of citations is accuracy. The shortcomings and difficulties in the current research were discussed, and the future development direction was expounded. To sum up, I think this paper deserves publication in Photonics after the authors consider the following revisions:

  1. There are a few typos to be corrected. For example, in line 967 of page 27, “CO2” should be replaced with “CO2”; In line 988, “L2” should be replaced with “L2”; In line 1134 of page 31, “dc-part”. In addition, there are many unrecognized words in manuscript that need to be modified. For example, in line 153 of page 4; in line 196 of page 6; in line 543 of page 15, etc.

  1. How is the author's review paper different from other review papers in this field?

  1. If possible, could the authors describe how to find these literatures?

Author Response

  Thanks a lot for such high evaluations on our manuscript. 

There are a few typos to be corrected. For example, in line 967 of page 27, “CO2” should be replaced with “CO2”; In line 988, “L2” should be replaced with “L2”; In line 1134 of page 31, “dc-part”. In addition, there are many unrecognized words in manuscript that need to be modified. For example, in line 153 of page 4; in line 196 of page 6; in line 543 of page 15, etc.

    The typos have been revised and all the unrecognized symbols mentioned above have been deleted.

How is the author's review paper different from other review papers in this field?

    Although there have been some of the outstanding review papers on fiber gratings (including the FBGs and the LPGs) and their applications, the review papers concentrated on topic of the phase-inserted fiber gratings have rarely been found in the past twenties years. It is the first time that the past and recent advances on the phase-inserted fiber gratings have been fully reviewed, to the best of our knowledge.

If possible, could the authors describe how to find these literatures?

   Two simple approaches have been adopted in order to find all the related literatures. The first one relies on the web searching, the second relies on tracing the referenced papers. We do read all the papers which are mentioned and cited in this review paper.       

    It is highly appreciated that the reviewer spent so much of his time to read our manuscript carefully. Thanks a lot for the invaluable comments.

Reviewer 2 Report

The manuscript is give a review of phase shift fiber grating and their applications in  Optical Filtering, Optical Signal Processing, and Optical Sensing. I think the review benefit the community of optical fiber devices. it is organized well and could be accepted as it is.

Author Response

The manuscript is given a review of phase shift fiber grating and their applications in optical filtering, optical signal processing, and optical sensing. I think the review benefits the community of optical fiber devices. It is organized well and could be accepted as it is.

    Thanks a lot for such high evaluation on this manuscript. It is highly appreciated that the reviewer spent so much of his time to read our manuscript carefully.